# Pluchea dioscoridis extract: A novel therapeutic approach for polycystic ovary syndrome targeting ovarian morphology, dopamine pathways, and neurobehavior in relation to its phytocomponents

Rasha Atta[1,2☯]*, Sahar Galal Gouda[3☯], Marwa Hussein Mohamed[4☯], Sherif M. Afifi[5‡], Mahmoud I. Nassar[6‡], Mohamed A. Farag[7‡], Abdelsamed I. Elshamy[6☯], Mohamed A. Zayed[8,9,10‡], Thamer Alqurashi[11‡], Shimaa Mohammad Yousof[1,8,10☯]

1 Medical Physiology Department, Faculty of Medicine, Suez Canal University, Ismailia, Egypt, 2 Center of Excellence in Molecular and Cellular Medicine (CEMCM), Faculty of Medicine, Suez Canal University, Ismailia, Egypt, 3 Department of Histology and Cell Biology, Faculty of Medicine, Suez Canal University, Ismailia, Egypt, 4 Department of Biochemistry and Molecular Biology, Faculty of Medicine, Suez Canal University, Ismailia, Egypt, 5 Department for Life Quality Studies, Rimini Campus, University of Bologna, Rimini, Italy, 6 Department of Natural Compounds Chemistry, National Research Centre, Giza, Egypt, 7 Pharmacognosy Department, Faculty of Pharmacy, Cairo University, Cairo, Egypt, 8 Medical Physiology, Faculty of Medicine, King Abdulaziz University, Rabigh, Saudi Arabia, 9 Medical Physiology Department, Faculty of Medicine, Menoufia University, Shibin Al Kawm, Egypt, 10 Neuroscience and Geroscience Unit, King Fahad Medical Research Centre, King Abdulaziz University, Jeddah, Saudi Arabia, 11 Faculty of Medicine in Rabigh, Pharmacology Department, King Abdulaziz University, Jeddah, Saudi Arabia

☯ These authors contributed equally to this work.
‡ SMA, MIN, MAF, MAZ and TA also contributed equally to this work.
* rasha_atta@med.suez.edu.eg

## Abstract

### Background

Polycystic ovarian syndrome (PCOS) is a prevalent endocrine condition associated with hormonal and metabolic abnormalities, as well as behavioral modifications.

### Aim

This research assesses the impact of the chemically profiled Pluchea dioscoridis ethanolic extract and metformin in a letrozole-induced polycystic ovary syndrome rat model.

### Methods

Thirty female Sprague Dawley rats were allocated into five groups: Control, P. dioscoridis (100 mg/kg), PCOS (letrozole, 1 mg/kg), PCOS + Metformin (300 mg/kg), and PCOS + P. dioscoridis (100 mg/kg), letrozole was given for 8 weeks followed by metformin or P. dioscoridis for 21 days. Behavioral assessments, hormone analyses,

**Data availability statement:** All relevant data are within the paper and its Supporting information files.

**Funding:** The author(s) received no specific funding for this work.

**Competing interests:** The authors have declared that no competing interests exist.

**Abbreviations:** FSH, Follicle-stimulating hormone; LH, Luteinizing hormone; MENA, Middle East and North Africa; PCOS, Polycystic Ovary Syndrome; SHBG, Sex Hormone Binding Globulin.

and dopamine quantifications in the brain were performed. Ovarian histology and immunohistochemical analysis were conducted.

## Results

Letrozole-induced PCOS resulted in heightened depression- and anxiety-like behaviors, along with significant hormonal imbalances compared to the control group ($P < 0.05\%$). Both *P. dioscoridis* and metformin therapies significantly ameliorated these changes, with *P. dioscoridis* demonstrating better efficacy. *P. dioscoridis* medication regulated luteinizing hormone (LH), follicle-stimulating hormone (FSH), testosterone, and estrogen levels, concurrently enhancing cerebral dopamine levels. Histological analysis revealed less cystic follicles and a reinstated normal ovarian architecture in rats treated with *P. dioscoridis* compared to PCOS group ($P < 0.05\%$). Several flavonoids, nitrogenous compounds and hydroxy cinnamic acid esters were detected in *P. dioscoridis* samples for the first time utilizing UPLC-ESI-MS analysis.

## Conclusion

The *P. dioscoridis* ethanolic extract exhibited potential medicinal properties that are equivalent to or surpass those of metformin for treating behavioral, hormonal, and ovarian structural abnormalities produced by PCOS. It significantly improved dopamine levels, hormonal balance, and ovarian histology, rendering it a suitable alternative or complementary treatment for PCOS.

## 1. Background

Polycystic ovary syndrome is an endocrine heterogeneous disease that is presented by a combination of excess androgen (hirsutism and acne), ovarian dysfunctions, and polycystic ovarian morphology [1]. The hormonal disturbance of luteinizing hormone (LH), follicular stimulating hormone (FSH), estrogen, and testosterone interferes with the normal menstrual cycles leading to menstrual irregularities and/or amenorrhea. PCOS is usually only detected when complications such as hair loss, acne, and infertility develop [2]. The multifactorial character of the disease results in the coexistence of multiple comorbidities including obesity, diabetes, infertility, cardiovascular and psychotic disorders [3].

PCOS has not only hormonal and metabolic effects but also significant psychological consequences such as depression, anxiety, obsessive-compulsive disorder, bipolar disorder, eating disorders, and reduction of sexual desire [4]. The link between depression and PCOS is complicated and bidirectional, with each condition affecting and exacerbating the other. PCOS patients suffer from depression due to hormonal imbalance and other physical presentations with impaired quality of life [5].

Animal models of depression, including learned helplessness and chronic mild stress (CMS) models, demonstrate substantial alterations in dopamine (DA) system functionality [6]. This encompasses modified dopamine receptor expression in

limbic areas, diminished synaptic dopamine release, lowered levels of the dopamine metabolite homovanillic acid, and decreased dopaminergic activity in the striatum [7–9]. Comparable patterns are noted in individuals with depression, indicating a persistent association between dopamine system downregulation and depressive states [6,8]. Anxiety is an undesirable condition characterized by an amplified reaction to situations. Principal cerebral regions, such as the amygdala, hippocampus, and frontal cortex, participate in its modulation. Neurotransmitter dysfunction, especially involving dopamine, significantly contributes to anxiety. Research indicates that the mesolimbic, mesocortical, and nigrostriatal dopaminergic systems, together with dopamine D1 and D2 receptors, play a vital role in the regulation of anxiety [10].

To date, there is no specific therapy for the treatment of PCOS but some interventional treatments are used to manage the symptoms. The treatment plan differs according to the symptoms and the underlying cause. It depends on the treatment of ovulatory disturbance, hyperandrogenism, and enhancing insulin resistance together with changing lifestyle [11].

Metformin is a biguanide drug used to treat insulin resistance and regulate menstrual abnormalities in PCOS patients [12]. Metformin acts by lowering the insulin and decrease androgen production by ameliorating CYP17 cytochrome action. In addition, it also increases the SHBG level with a further decrease in the level of free testosterone [13]. Additionally, it improves the lipid profile of PCOS women [14]. The effect of metformin on dopamine and its related release structures in the brain is controversial and in need of further studies [15].

*P. dioscoridis* (L.) DC., also known as *Conyza dioscoridis* (L.) Desf. is a widespread desert plant found in the Sinai Peninsula, Western Desert, Eastern Desert, and Mediterranean coast [16]. This plant has been used in traditional medicine to treat a variety of conditions, including colds, rheumatic aches, ulcers, colic, carminative symptoms, and pediatric epilepsy [16]. Numerous studies have reported that this plant's various extracts have a diuretic effect in addition to several strong biological properties, including reducing inflammation, gastroprotection, hypoglycemic, antinociceptive, antipyretic, antimicrobial, and antioxidant effects [17–19]. Several metabolites were characterized from this plant including flavonoids, phenolic acids, steroids, and triterpenes [19] in addition to essential oil [20].

To our knowledge, the therapeutic potential of *P. dioscoridis* in the management of PCOS and its related psychosocial effects remains unexplored. This study seeks to examine the possible beneficial effects of *P. dioscoridis* ethanolic extract on ovarian morphological alterations and psychological consequences in a rat model of PCOS in relation to its characterized bioactive phytochemical metabolites. Furthermore, the dopamine-related mechanisms at several levels were assessed, including receptor expression and dopamine degradation by monoamine oxidase, within the framework of PCOS. Also, the efficacy of *P. dioscoridis* EtOH extract in comparison to metformin, a commonly utilized therapeutic medication, was evaluated to deliver a thorough assessment of the plant's potential as an alternate medication choice.

## 2. Methods

### 2.1. Plant materials, authentication, and extraction

The fresh and wholesome aerial parts of *P. dioscoridis* were gathered early on May 4, 2022, between 4–6 AM, from Wadi el-Natroun (30°35′N 30°20′E), along the Cairo–Alexandria desert route in Egypt. Professor of Taxonomy at Mansoura University, Prof. Dr. Ahmed M. Abd ElGawad, gathered and verified the plant material obtained. A voucher specimen (Mans-PL-022-Gpx-4481) has been placed in the Mansoura University Herbarium in Egypt. Following a period of shade drying, the plant parts were crushed into a finely ground form and then placed aside to undergo additional extraction.

The air-dried plant materials (820 gm) were macerated in 4 liters of a 7:3 mixture of ethanol and distilled water for 5 days at room temperature then filtered. After completing three rounds of this process, all the liquor extracts were gathered and dried under vacuum, producing 34.2 g of black gum that was kept refrigerated at 4 °C until additional testing was done [21].

#### 2.1.1. UPLC-ESI -MS chemical analysis.
The UPLC-ESI-QTOF-MS techniques of *P. dioscoridis* EtOH extract were implemented in compliance with the guidelines given by Yousof et al. (2024) and Kabbash (2023) [22,23].

## 2.2. Chemicals and drugs

Carboxymethylcellulose (SIGMA, USA), letrozole (Femara ® 2.5 mg, Novartis Pharma AG, Basel, Switzerland), metformin (Cedophage ® 500 mg, CID pharma, Egypt).

## 2.3. Experimental design: (Fig 1)

**2.3.1. Animals.** A total of 30 female Sprague Dawley rats 6–8 weeks old (weighing 200–250 g) were provided by Moustafa Rashed Company for experimental animals (Giza, Egypt). The rats were housed in a hygienic laboratory environment and a normal light-dark cycle. They were kept for 1 week acclimatization period. The experiment was approved by the research ethics committee, Faculty of Medicine, Suez Canal University, Egypt (No. 4955#).

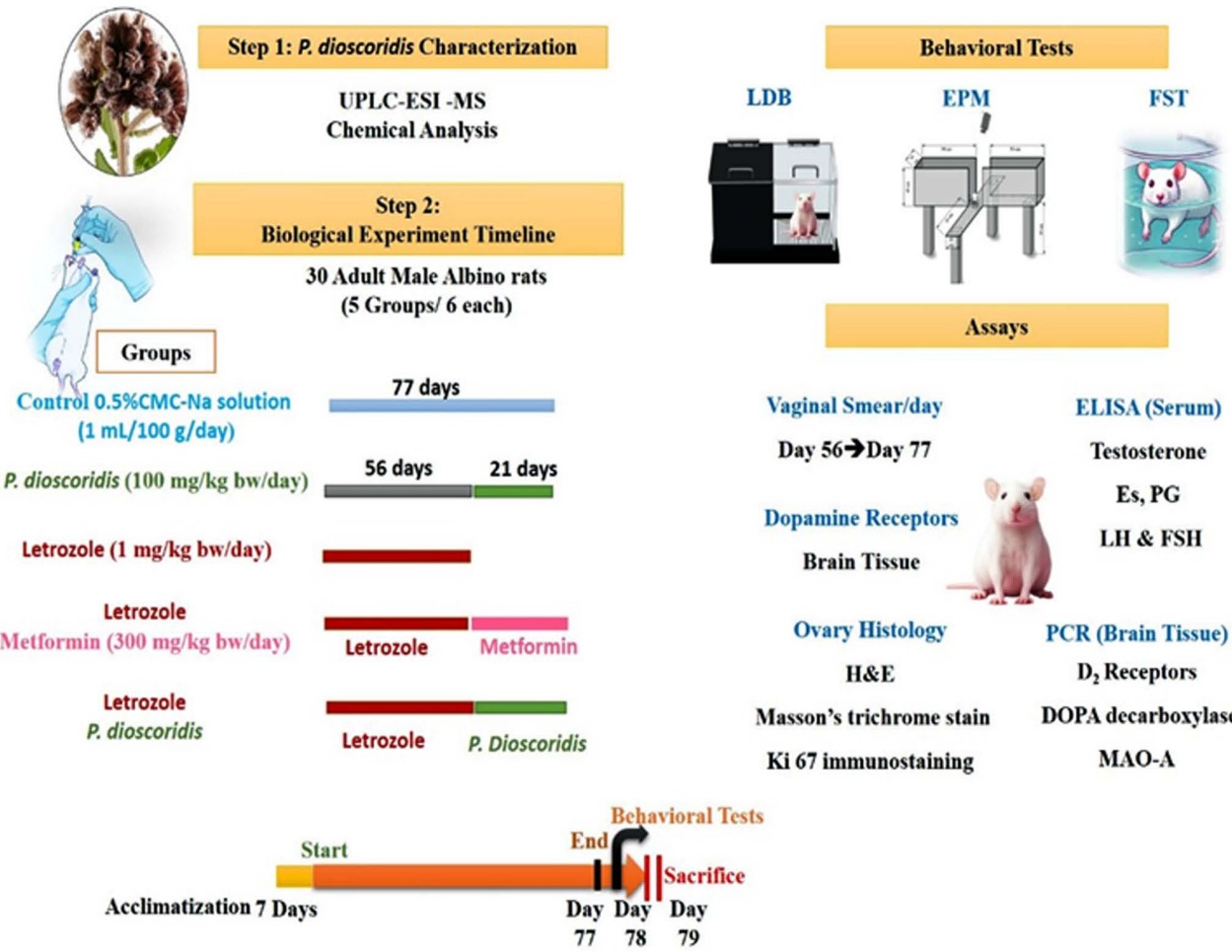

**Fig 1. Experimental Plan for Assessing *P. dioscoridis's* Impact in a PCOS Rat Model.** The figure describes a two-step experimental procedure that involves a biological experiment after *P. dioscoridis* characterization. Treatments were administered to five experimental groups: Control (0.5% CMC-Na), *P. dioscoridis* (100 mg/kg), letrozole (1 mg/kg), letrozole + metformin (300 mg/kg) and letrozole + *P. dioscoridis*. Towards the end of the study, behavioral tests including the forced swim test (FST), elevated plus maze (EPM), and light-dark box (LDB) were carried out. Daily vaginal smears, brain tissue PCR, ovary histology, serum ELISA, and dopamine receptor analysis were conducted. The timeline emphasizes sacrifice points, behavioral evaluations, and treatment duration.

**2.3.2. Preparation of a PCOS rat model and treatment.** The rats were randomly allocated into five groups (6 rats in each group). The control group was given a 0.5% carboxymethylcellulose (CMC)-Na solution (1 mL/100 g/day; SIGMA, USA) through intragastric gavage for a duration of 11 weeks. The *P. dioscoridis* group: the rats were administered *P. dioscoridis* EtOH extract (100 mg/kg body weight) orally during the last 21 days of the experiment [24]. The PCOS: the rats received letrozole (1 mg/kg/day) dissolved in 1% (w/w) CMC by intragastric gavage daily for 8 weeks to develop PCOS [25,26], the duration of PCOS induction was based on a pilot study and on a study done by Zheng et al., 2021 [25].

The PCOS + Metformin: the rats were administered metformin (300 mg/kg body weight) orally for 21 days subsequent to PCOS induction [27], whereas the PCOS + *P. dioscoridis* group: rats received *P. dioscoridis* extract (100 mg/kg body weight) orally for 21 days following PCOS induction [28]. The choice of method of administration and duration of metformin treatment was based on similar ones in respect to *P. dioscoridis* administration to standardize the treatment.

**2.3.3. Vaginal smear preparation.** The changes in oestrus cycles were monitored with a light microscope via vaginal smears. Vaginal smears were collected each day from the 8ᵗʰ week to the end of the experiment. The smears were fixed with 10% formalin and stained with hematoxylin and eosin. The oestrus cycle was determined based on the previous literature [29].

**2.3.4. Neurobehavioral tests.** At the end of the experiment, all animals were subjected to neurobehavioral tests. Rats were assessed for the presence of depression and anxiety.

**2.3.4.1. Forced Swimming Test (FST):** One of the reliable techniques for evaluating animal depression is the FST. Each rat was situated in a cylindrical tank (diameter: 25 cm, height: 50 cm) that was filled with water (23–25 °C). Each animal was permitted to freely swim in the water and was videotaped for six min. For further offline analysis. Immobility was defined as floating without making any movements other than those required to maintain the nose above water, and this time was noted as an index. Depression is indicated by a high immobility time [30].

**2.3.4.2. Elevated plus maze (EPM):** The rat EPM is a medium-density fiberboard with a matte black acrylic surface, consisting of four arms. The open arms have a railing, and the maze is elevated 50 cm off the floor. The maze was cleaned and dried before use, and a video-tracking was used for further offline analysis. The rodent is placed in the maze facing the same open arm, and the video-tracking system records the number of entries made by the rodent onto the open and closed arms and the time spent on the open arm and closed arm. The observer must avoid unnecessary movements and noise making. At the end of the 5-min. test, the rodent is removed from the maze and placed back inside its home cage [31].

**2.3.4.3. Light dark box:** The device consisted of two compartments, one illuminated large (30 x 33 x 30 cm) and one tiny dark (20 x 33 x 30 cm), with a connecting gate (4 x 7 cm) in between. Each rat was placed in the center of the light compartment facing the gate. Each rat was videotaped for 5 min. duration. The time spent, and number of entries to the light compartment, and frequency rearing (exploratory activity) were recorded [32].

**2.3.5. Plasma assays and tissue processing.** At the end of the experiment, rats were anesthetized by intraperitoneal injection of 2% sodium pentobarbital (0.2 mL/100 g), and serum samples were collected after intracardiac puncture and stored at −20 °C. Both brain and right ovaries were collected and stored at −20 °C for further biochemical, while the left ovaries were fixed in Bouin fixative for histopathological studies.

**2.3.6. Measurement of hormonal levels.** Serum Levels of LH, FSH, estrogen, progesterone, and testosterone were measured by Rat LH ELISA Kit (My Biosource, San Diego, USA, cat lot.: MBS2509833), Rat FSH ELISA Kit (My Biosource, San Diego, USA, cat lot.: MBS017508), Rat Estrogen ELISA Kit (My Biosource, San Diego, USA, cat lot.:MBS703614), Rat Progesterone ELISA Kit (Cusabio, Houston, USA, cat lot.: CSB-E07282r) and Rat Testosterone ELISA Kit ((My Biosource, San Diego, USA, cat lot.: MBS262661), respectively. Rat Dopamine (DA) Elisa kit My Biosource, San Diego, USA, cat lot.: MBS7214676) was used to measure dopamine level in brain homgenate, and the absorbance was then measured at 450 nm using a microplate spectrophotometer (Multiskan GO, Thermo, Waltham, MA, USA).

 

**2.3.7. Histopathological analysis and immunohistochemistry.** Left ovaries were placed in Bouin fixative for 24 hours and then transferred to an ascending series of ethanol for dehydration, cleared in xylene, impregnated in paraffin wax, and cut into 5 µm-thick paraffin sections for histological and immunohistochemical techniques. Histologically, sections were stained with routine hematoxylin and eosin (H&E) stain for the general architecture of the ovary and Masson's trichrome stain for the collagen fibers in the stroma, theca externa, and corpora lutea. The sections of the ovary were examined and imaged at a magnification of 100x and 200x using a Leica DM1000 microscope connected to a digital camera, Leica DFC425, Germany. The following parameters were quantitatively evaluated in the ovary: the mean thickness (µm) of the granulosa layer and the mean thickness (µm) of the theca layer in H&E-stained sections (at magnification 200x), the mean count of secondary (antral) follicles and the mean count of atretic (degenerated) follicles per ovary in H&E-stained sections (at magnification 100x), the mean count of corpora lutea, and the mean count of cystic follicles per ovary in H&E-stained sections (at magnification 100x), and the mean color area percentage of greenish collagen in the ovary (Masson's trichrome stain at magnification 100x). Quantitative assessment was done using ImageJ software (NIH, USA). Eight non-overlapping fields per section and 3 sections/animal were evaluated.

**2.3.7.1. Ki 67 immunostaining:** For ki67 immunostaining, rabbit monoclonal primary antibody (ki67 rabbit mAb, dilution 1:200, ABclonal, USA, Catalog No. A20018) was used. Five µm-thick paraffin sections were cut and gathered on positively charged slides. They were dried in an oven for an entire night at 58 °C for optimal adherence. They were then deparaffinized, rehydrated, and put in 0.01 M citrate buffer (pH = 6.0) and microwave-irradiated to retrieve the antigen [33]. Next, sections were put in a humid chamber and allowed to incubate for 5 min. in a peroxidase\AP blocker, followed by 60 min. in Ki67 primary antibody incubation. For 45 min, the horseradish peroxidase label (HRP) was applied, followed by 10 min. of 3, 3'-DAP chromogen (Mouse/Rabbit PolyDetector, Bio SB, USA). Meyer's hematoxylin was used as a counterstain for 30 seconds on the sections [34]. PBS was used in place of the primary antibody as the negative control. A section of the small intestine served as the positive control. Morphometry: Using Image J software (NIH, USA), the mean color area percentage of Ki67 immunostaining in the ovary was assessed. Twenty captures per group were evaluated.

**2.3.8. Assessment of dopamine receptor 2, monoamine oxidase A (MAO-A), and DOPA decarboxylase genes by quantitative real-time PCR.** Expression of dopamine receptor 2, MAO-A, and DOPA decarboxylase genes were analyzed using quantitative real-time PCR (qRT-PCR, Applied Biosystems, Waltham, MA, USA).

**2.3.8.1. Total RNA extraction:** Total RNA extraction was done from brain tissue homogenate samples using TRIzol reagents according to the manufacturer's protocol (Applied Biosystems), followed by RNase-free DNase 1 treatment to remove any DNA contamination according to the manufacturer's guidelines (Applied Biosystems, Waltham, MA, USA).

**2.3.8.2. RNA assessment:** The extracted RNA was evaluated for integrity and concentration using a Nanodrop-1000 spectrophotometer (Nanodrop Technologies, Wilmington, DE, USA). RNA purity was measured by determining the optical density (OD) of the RNA extracts at 260 nm and 280 nm.

**2.3.8.3. cDNA synthesis by reverse transcription:** cDNA was synthesized from isolated RNA using a reverse transcription Kit (Applied Biosystem) according to the manufacturer's guidelines, on Veriti™ 96-Well Fast Thermal Cycler (Cat#4375305, Applied Biosystems, Waltham, MA, USA).

**2.3.8.4. Real-time quantitative PCR conditions:** Targeted genes' mRNA expression level was quantified compared to β-Actin as a housekeeping gene using qRT-PCR on StepOne Plus™ Real-Time PCR System (Cat#4376600, Applied Biosystems, Waltham, MA, USA) in combination with continuous SYBR Green detection (Applied Biosystems). Real-time PCR was performed in a 10 µl reaction volume containing 1µl cDNA, 5µl SYBR Green PCR Master Mix, 1µl each primer, and 2µl distilled water. The program was set as follows: initial polymerase activation at 95ºC for 10 min, followed by 40 cycles of denaturation at 95 ºC for 15s, annealing at 60 ºC for 30 s, and extension at 72 ºC for 60s. Relative gene expression was assessed using the following primer pairs **shown in** Table 1 [35–37].

**Table 1. PCR gene expression for dopamine receptor 2, MAO-A and DOPA decarboxylase.**

| Targeted gene | Primer Sequence | Reference |
|---|---|---|
| β-Actin | F: 5' AGGCCCCTCTGAACCCTAAG 3'<br>R: 5' GGAGCGCGTAACCCTCATAG 3' | [35] |
| Dopamine receptor 2 | F:5' TGAACCTGTGTGCCATCAGCA 3'<br>R:5' TTGGCTCTGAAAGCTCGACTG 3' | [35] |
| MAO-A | F: 5' GCCAGGAACGGAAATTTGTA 3'<br>R: 5' TCTCAGGTGGAAGCTCTGGT 3' | [36] |
| DOPA decarboxylase | F: 5' TTC GAA AGC ACG TGA AGC TG 3'<br>R: 5' CCG TGC AAATTT CAA AGC G 3' | [37] |

## 2.4. Statistical analysis

SPSS version 22 was utilized for statistical analysis. The data was presented as mean values ± SEM. The significance of the data was assessed using a one-way ANOVA test, with a post hoc Bonferroni test applied. A *p*-value of less than 0.05 was considered statistically significant.

## 3. Result

### 3.1. Chemical profile of *P. dioscoridis* EtOH extract

The chemical makeup of P. dioscoridis EtOH extract was investigated utilizing UPLC-ESI-MS analysis (Fig 2) in -ve and +ve modes. The proposed identification of metabolites was carried out based on the comparison of molecular ion peaks

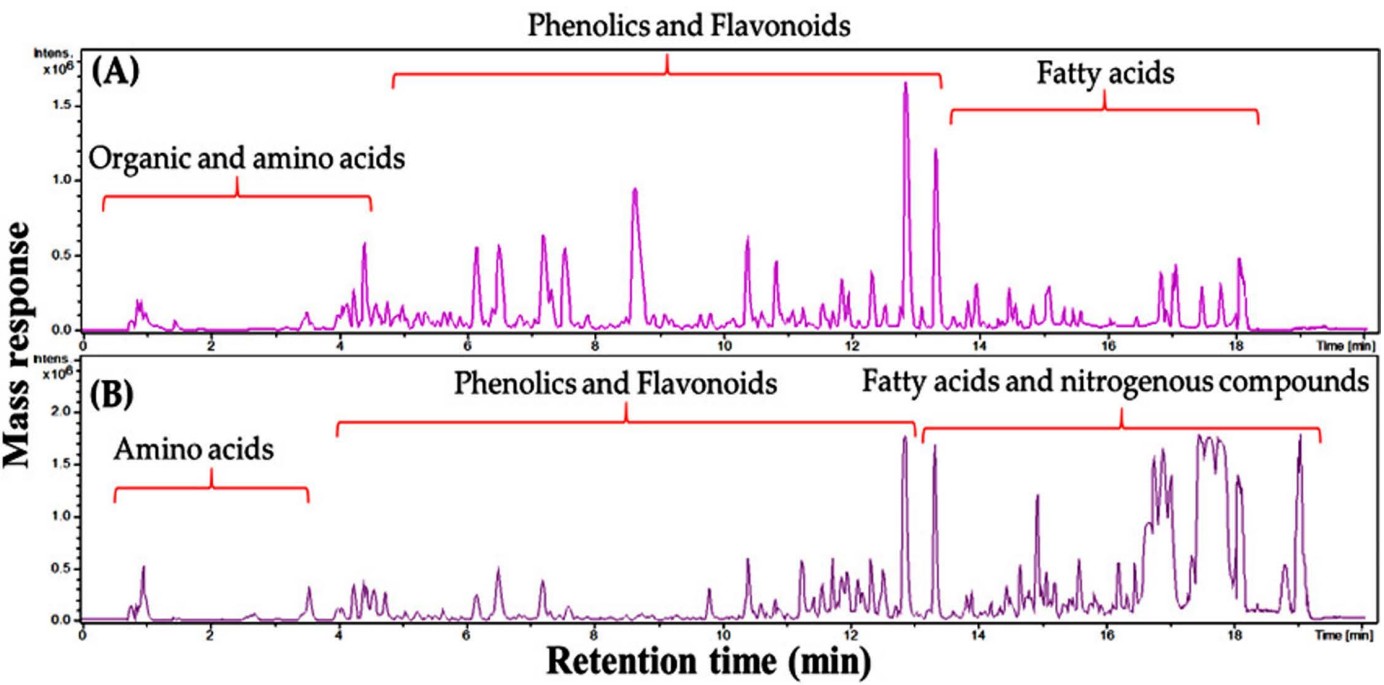

**Fig 2. Representative UPLC-MS-MS chromatogram of *P. dioscoridis* ethanolic extract performed in (A) negative and (B) positive ionization mode.**

and mass ion fragments to the documented data. *P. dioscoridis* analytes, 56 in total, encompassed 26 flavonoids, 5 fatty acids, and 8 phenolic acid derivatives besides nitrogenous compounds and terpenoids as depicted in Table 2. *P. dioscoridis* exhibited strong antioxidant potential due to multiple hydroxycinnamic acid esters and a rich flavonoid content, including quercetin and luteolin—both linked to neuroprotective effects [16,38]. Nitrogenous compounds such as sphingosines and vitamin B5 were also identified, highlighting its metabolic relevance [39].

### 3.2. Effect of metformin and *P. dioscoridis* extract on estrous cycle

Disturbance of the normal estrous cycle was confirmed in the PCOS group. Letrozole induced dominant diestrus stage revealing irregularity of the estrous cycle as compared to the control group (Figs 3 and 4). Both metformin and *P. dioscoridis* extract treatments restored regular phases of the estrous cycle when compared with control and PCOS groups (Fig 4).

### 3.3. Effect of metformin and *P. dioscoridis* extract on depressive-like behavior

Forced swimming test data revealed a significant increase in immobility time in rats treated with letrozole compared to the control group ($P < 0.05$). Treatment with metformin or *P. dioscoridis* showed significant improvement in immobility time compared to the PCOS group ($P < 0.05$), with *P. dioscoridis* showing better results (Fig 5A).

### 3.4. Effect of metformin and *P. dioscoridis* extract on anxiety-like behavior

Light-dark box assessment revealed that letrozole administration for 8 weeks showed a significant decrease in the time spent in the light box ($P < 0.05$) (Fig 5A). PCOS group with *P. dioscoridis* showed preserved time spent in the light box near the normal control group when compared to the PCOS group ($P < 0.05$), referring to the anxiolytic effect of *P. dioscoridis* [59], (Fig 5B). Regarding the frequency of rearing, the PCOS group showed a significant decrease in comparison to the control group. Whilst *P. dioscoridis* preserved the frequency of rearing near normal when compared to the PCOS group ($P < 0.05$), referring to increased exploration behavior (Fig 5C). However, there were not any significant differences among groups in the number of entries to light box ($P > 0.05$) (Fig 5D).

An elevated plus maze was used also to measure anxiety like-behavior and revealed a significant increase in the time spent in the closed arm, with a significant decrease in the time spent in the open arm in both PCOS and metformin groups compared to the control group ($P < 0.05$), pointing to increased anxiety level [60] (Fig 5E). *P. dioscoridis* + PCOS group showed results similar to the control group in time spent in the open and closed arms (Fig 5E).

### 3.5. Effect of metformin and *P. dioscoridis* extract on serum LH, FSH, testosterone, progesterone and estrogen values

Letrozole administration in the PCOS group for 8 weeks resulted in a significant increase in serum LH, serum FSH, and serum testosterone compared to both control and *P. dioscoridis* groups, $P < 0.05$ (Fig 6A, 6B, 6C) ($P = 0.05$) (Fig 6C). In addition to a significant decrease in both serum progesterone and estrogen compared to control and *P. dioscoridis* groups $P < 0.05$ (Fig 6D, 6E).

Treatment with either metformin or *P. dioscoridis* extracts after PCOS induction showed a significant improvement in sex hormonal profile compared to the PCOS group with more significant improvement in the *P. dioscoridis* extract group compared to the metformin group. Serum LH and FSH were significantly decreased in both metformin and *P. dioscoridis* extract-treated groups compared to the PCOS group, $P < 0.05$(Fig 6A, 6B).

Hyperandrogenemia was significantly improved in both treated groups with more significant improvement in *P. dioscoridis* extract-treated group compared to metformin, $P < 0.05$ (Fig 6C). Serum progesterone and estrogen were significantly increased in the *P. dioscoridis* extract-treated group compared to the metformin group, $P < 0.05$ (Fig 6D, 6E)

**Table 2. Annotated metabolites in *P. dioscoridis* EtOH extract *via* UPLC-MS-MS.**

| No. | RT [min] | Name | Class | Precursor | Formula | Error (ppm) | Fragments | Ref. |
|---|---|---|---|---|---|---|---|---|
| 1 | 0.967 | Gluconic acid | Sugar acid | 195.0511 | $C_6H_{11}O_7^-$ | 0.3 | 177, 151, 129, 99, 87, 75 | [40] |
| 2 | 1.002 | Quinic acid | Organic acid | 191.0557 | $C_7H_{11}O_6^-$ | −2.1 | 127 | [41] |
| 3 | 2.679 | Phenylalanine | Amino acid | 166.0864 | $C_9H_{12}NO_2^+$ | 0.8 | 131, 120, 107, 103 | [42] |
| 4 | 3.065 | Dihydroxy-dimethyl-oxobutyl-Alanine (Pantothenic acid) | Nitogenous compound | 220.1183 | $C_9H_{18}NO_5^+$ | 1.5 | 202, 184, 116, 90 | [43] |
| 5 | 3.275 | Xanthurenic acid | Quinoline | 206.0446 | $C_{10}H_8NO_4^+$ | −0.8 | 178, 160, 132 | [44] |
| 6 | 3.983 | Caffeoyl-quinic acid | Phenolics | 353.0874 | $C_{16}H_{17}O_9^-$ | −1.1 | 191 | [45] |
| 7 | 4.011 | Glutamylphenylalanine | Amino acid | 295.1278 | $C_{14}H_{19}N_2O_5^+$ | −3.5 | 278, 261, 232, 166 | |
| 8 | 4.253 | Kaempferol-*C*-deoxyhexosyl-*C*-hexoside | Flavonoid | 595.1664 | $C_{27}H_{31}O_{15}^+$ | 1.1 | 577, 559, 475, 457 | [46] |
| 9 | 4.396 | Caffeic acid | Phenolics | 179.0344 | $C_9H_7O_4^-$ | −3.2 | 135 | [47] |
| 10 | 4.459 | Apigenin-*C*-pentosyl-*C*-hexoside (Isoschaftoside) | Flavonoid | 565.1553 | $C_{26}H_{29}O_{14}^+$ | 0.2 | 547, 529, 505, 499, 475, 445, 427, 415, 409, 385, 356 | [48] |
| 11 | 4.665 | Luteolin-*C*-hexoside | Flavonoid | 449.1074 | $C_{21}H_{21}O_{11}^+$ | −0.9 | 431, 413, 383, 353,329, 299 | [46] |
| 12 | 4.82 | *O*-Caffeoylquinic acid methyl ester (Methyl chlorogenate) | Phenolics | 369.1184 | $C_{17}H_{21}O_9^+$ | 1.1 | 177, 163 | |
| 13 | 5.044 | Trihydroxy-flavone-*C*-hexoside (Isovitexin) | Flavonoid | 433.1133 | $C_{21}H_{21}O_{10}^+$ | 0.8 | 415, 313, 283 | [48] |
| 14 | 5.267 | Pentahydroxyflavone-*O*-deoxyhexosyl-hexoside (Rutin) | Flavonoid | 611.1609 | $C_{27}H_{31}O_{16}^+$ | 0.3 | 465, 303 | [46] |
| 15 | 5.372 | Pentahydroxy-methoxyflavone-*O*-deoxyhexosyl-hexoside | Flavonoid | 641.1715 | $C_{28}H_{33}O_{17}^+$ | 0.4 | 495, 333 | |
| 16 | 5.65 | Quercetin-*O*-hexoside (isoquercitrin) | Flavonoid | 465.1025 | $C_{21}H_{21}O_{12}^+$ | −0.5 | 303 | [42] |
| 17 | 5.76 | Kaempferol-*O*-glucuronide | Flavonoid | 463.0872 | $C_{21}H_{19}O_{12}^+$ | 0.2 | 287 | [49] |
| 18 | 5.783 | Kaempferol-*O*-deoxyhexosyl-hexoside | Flavonoid | 595.166 | $C_{27}H_{31}O_{15}^+$ | 0.4 | 449, 287 | [46] |
| 19 | 6.029 | Tetrahydroxy-methoxy-flavone-*O*-deoxyhexosyl-deoxyhexoside (Narcissoside) | Flavonoid | 625.177 | $C_{28}H_{33}O_{16}^+$ | 1.1 | 479, 317, 85 | |
| 20 | 6.138 | Dicaffeoylquinic acid (Cynarine) | Phenolics | 515.1197 | $C_{25}H_{23}O_{12}^-$ | 0.3 | 353, 191 | [50] |
| 21 | 6.164 | Dicaffeoylquinic acid (dimer) | Phenolics | 1033.2631 | $C_{50}H_{49}O_{24}^+$ | 2.1 | 517, 499, 355, 337, 163 | |
| 22 | 6.281 | Cimicifugic acid | Phenolics | 449.1076 | $C_{21}H_{21}O_{11}^+$ | −0.5 | 341, 303, 287, 273, 255, 193 | [51] |
| 23 | 7.18 | Cynarine isomer | Phenolics | 517.1337 | $C_{25}H_{25}O_{12}^+$ | −0.6 | 163 | |
| 24 | 7.214 | Dicaffeoylquinic acid lactone | Phenolics | 499.1231 | $C_{25}H_{23}O_{11}^+$ | −0.7 | 337, 175 | [52] |
| 25 | 7.584 | Trihydroxy-methoxy-flavone-*O*-hexoside | Flavonoid | 463.1229 | $C_{22}H_{23}O_{11}^+$ | −1.2 | 301, 201 | [46] |
| 26 | 9.455 | Trihydroxy-trimethoxy-flavone-*O*-hexoside | Flavonoid | 523.1441 | $C_{24}H_{27}O_{13}^+$ | −0.9 | 361 | |
| 27 | 9.77 | Trihydroxyflavone-*O*-dihexoside | Flavonoid | 593.1528 | $C_{27}H_{29}O_{15}^-$ | 2.7 | 431, 269 | [46] |
| 28 | 9.82 | Hypnophilin | Terpenoid | 249.1481 | $C_{15}H_{21}O_3^+$ | −1.6 | 231, 213, 203, 185, 177, 175 | |
| 29 | 10.051 | Dihydroxy-dimethoxyflavone-*O*-hexoside (Cirsimarin) | Flavonoid | 477.1372 | $C_{23}H_{25}O_{11}^+$ | −4 | 315 | |
| 30 | 10.287 | Tetrahydroxyflavone (Luteolin) | Flavonoid | 285.0415 | $C_{15}H_9O_6^-$ | 3.6 | 241, 175 | |
| 31 | 10.314 | Pentahydroxyflavone (Quercetin) | Flavonoid | 301.0356 | $C_{15}H_9O_7^-$ | 0.7 | 179, 151 | [53] |
| 32 | 10.475 | Tricin-*O*-sulfate | Flavonoid | 409.0233 | $C_{17}H_{13}O_{10}S^-$ | −0.4 | 329 | |
| 33 | 11.225 | Trihydroxy-trimethoxyflavone (Jaceidin) | Flavonoid | 359.076 | $C_{18}H_{15}O_8^-$ | −3.4 | 344, 329 | |
| 34 | 11.328 | Jaceidin-*O*-sulfate | Flavonoid | 439.0319 | $C_{18}H_{15}O_{11}S^-$ | −4.9 | 359 | |
| 35 | 11.392 | Trihydroxy-methoxy-flavone | Flavonoid | 301.07 | $C_{16}H_{13}O_6^+$ | −2.2 | 286, 283, 255, 241, 137 | [48] |
| 36 | 11.415 | Costunolide | Terpenoid | 233.153 | $C_{15}H_{21}O_2^+$ | −7.3 | 215, 187, 177, 159 | |
| 37 | 11.545 | Trihydroxy-dimethoxyflavone (Tricin) | Flavonoid | 329.0663 | $C_{17}H_{13}O_7^-$ | −1.1 | 314, 299 | |

*(Continued)*

**Table 2.** (Continued)

| No. | RT [min] | Name | Class | Precursor | Formula | Error (ppm) | Fragments | Ref. |
|---|---|---|---|---|---|---|---|---|
| 38 | 12.021 | Dihydroxy-tetramethoxyflavone (Casticin) | Flavonoid | 375.1061 | $C_{19}H_{19}O_8^+$ | −3.5 | 360, 342, 317, 231, 215, 179 | [54] |
| 39 | 12.254 | Dihydroxy-tetramethoxyflavone-sulfate | Flavonoid | 453.0484 | $C_{19}H_{17}O_{11}S^-$ | −2.8 | 373 | |
| 40 | 12.302 | Dihydroxy-dimethoxyflavone (Velutin) | Flavonoid | 313.0706 | $C_{17}H_{13}O_6^-$ | −3.7 | 298, 280 | [55] |
| 41 | 12.335 | Dihydroxy-dimethoxyflavone (Velutin) | Flavonoid | 315.0856 | $C_{17}H_{15}O_6^+$ | −2.2 | 300, 282, 257, 201, 187 | [55] |
| 42 | 12.509 | Dihydroxy-trimethoxyflavone | Flavonoid | 343.0815 | $C_{18}H_{15}O_7^-$ | −2.4 | 328, 313 | [46] |
| 43 | 12.535 | Hydroxy-oxooctadeca-trienoic acid | Fatty acid | 307.1911 | $C_{18}H_{29}O_4^-$ | −1.2 | 289, 235, 185, 121, 97 | |
| 44 | 13.138 | Sphingosine | Sphingo-lipid | 300.2887 | $C_{18}H_{38}NO_2^+$ | −3.3 | 282, 264, 252 | [56] |
| 45 | 13.319 | Atractylenolide III (dimer) [2M-H] | Naphtho-furan | 495.274 | $C_{30}H_{39}O_6^-$ | −2.4 | 247 | |
| 46 | 14.463 | Octadeca-tetraenoic acid | Fatty acid | 277.2149 | $C_{18}H_{29}O_2^+$ | −4.7 | 259, 135, 121 | [45] |
| 47 | 15.034 | Hydroxyoctadecadienoic acid | Fatty acid | 295.2276 | $C_{18}H_{31}O_3^-$ | −0.9 | 277, 195 | |
| 48 | 15.13 | Diazinon | Nitogenous compound | 305.1071 | $C_{12}H_{22}N_2O_3PS^+$ | −4 | 277, 249, 181, 169, 153 | [57] |
| 49 | 15.223 | Palmitoyl-glycero-phosphoethanolamine | Phoslipid | 454.2925 | $C_{21}H_{45}NO_7P^+$ | −0.6 | 436, 393, 313, 282, 216 | [58] |
| 50 | 15.457 | Hydroxy-octadecatrienoic acid | Fatty acid | 293.2123 | $C_{18}H_{29}O_3^-$ | 0.2 | 275, 171 | |
| 51 | 15.752 | Hydroxy-octadecatrienoic acid (dimer) [2M-H] | Fatty acid | 587.4309 | $C_{36}H_{59}O_6^-$ | −1.3 | 293 | |
| 52 | 16 | Pygenic acid B | lipid | 487.3418 | $C_{30}H_{47}O_5^-$ | −2.2 | – | |
| 53 | 16.319 | Linoleamide | Nitogenous compound | 324.2888 | $C_{18}H_{34}NO^+$ | −2.7 | 307, 263, 245 | |
| 54 | 16.439 | Icosa-tetraenoic acid (Arachidonic acid) | Fatty acid | 305.2465 | $C_{20}H_{33}O_2^+$ | −3.2 | 287, 259, 163, 149 | |
| 55 | 16.826 | Docos-enamide (Erucamide) | Fatty amide | 338.3406 | $C_{22}H_{44}NO^+$ | −3.3 | 321, 303 | |
| 56 | 17.236 | Epoxylanosta-dien-one (Cornusalterin L) | Terpenoid | 439.3557 | $C_{30}H_{47}O_2^+$ | −3 | 249, 203, 191 | |

### 3.6. Effect of metformin and *P. dioscoridis* extract on brain dopamine level

Fig 6F shows brain dopamine levels at the end of the experiment. There were significantly lower levels in dopamine in letrozole induced PCOS group compared to the control group ($P < 0.05$). Both metformin and *P. dioscoridis* extract-treated groups showed significantly elevated brain dopamine levels compared to the PCOS group ($P < 0.05$).

### 3.7. Correlation between behavioral measures and brain dopamine levels in studied groups

Table 3 shows Pearson correlation coefficients between immobility time in the forced Swimming Test, time spent in the closed arm and open arm of the elevated plus maze, and dopamine levels. Both tests' behavioral measures appear to be significantly correlated with dopamine. Lower immobility time (less depressive-like behavior), less time in closed arms, and increased time in open arms (lower anxiety-like behavior) are all correlated with higher dopamine levels.

### 3.8. Correlation between sex hormonal profile and brain dopamine levels in studied groups

Table 4 shows Pearson correlation coefficients between sex hormonal profile and dopamine levels. There were significant negative correlations between dopamine and LH, FSH and testosterone levels. There were significant positive correlations between dopamine, and progesterone and estrogen levels. Higher dopamine levels are associated with reduced testosterone, LH, and FSH, whereas associated with increased estrogen, and progesterone.

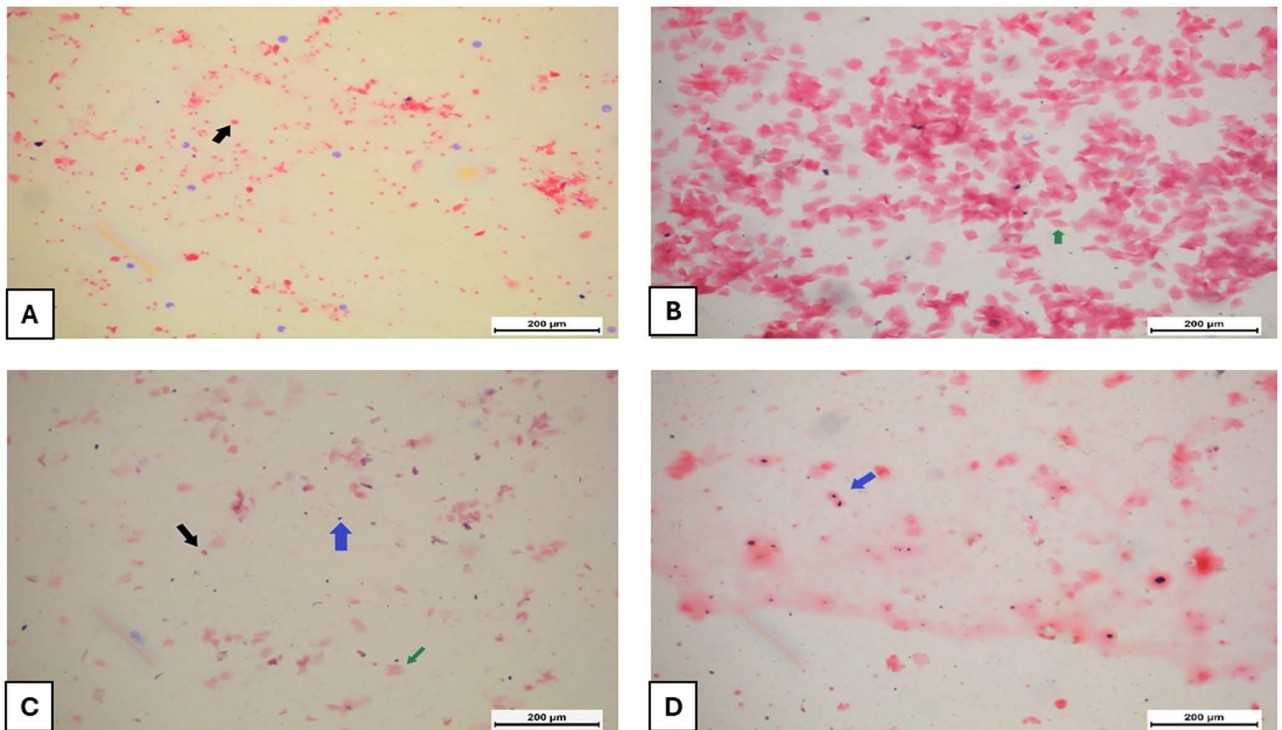

**Fig 3. Vaginal smear of normal control group showing the four stages of the estrous cycle.** (A) Proestrous phase containing predominantly small nucleated epithelial cells (black arrow). (B) Estrous phase characterized by anucleated keratinized epithelial cells (green arrow). (C) Metestrous phase: in this phase, leukocytes (blue arrow), small nucleated epithelial cells, and anucleated keratinized epithelial cells are present. (D) Diestrous phase mainly comprises of leukocytes (blue arrow).

### 3.9. Effect of metformin and *P. dioscoridis* extract on histological results

**3.9.1. Hematoxylin & Eosin (H&E) stain.** The ***control group*** showed normal histological structure of the ovary with superficial cortex and deep medulla. The medulla is composed of a highly vascularized stroma of loose connective tissue. The cortex had various ovarian follicles; primordial (PF), unilaminar primary (UF), multilaminar primary (MF), secondary (antral) (SF), and mature Graffian (GF) follicle. The primordial follicle (PF) had a large oocyte surrounded with a single layer of flattened follicular cells. The unilaminar primary (UF) follicle had a single layer of cuboidal cells surrounding the central oocyte. The multilaminar primary (MF) follicle had a large oocyte enclosed within the zona pellucida (ZP) layer and surrounded by several rows of cuboidal granulosa (G) cells and an outer sheath of theca folliculi (TF). The antral (secondary) follicle (SF) had spaces (antrum) between granulosa (G) cells contains liquor folliculi, and was ensheathed with inner cellular theca interna (TI) and outer fibrous theca externa (TE). The mature follicle (GF) had large antrum with liquor folliculi. The oocyte enclosed within the zona pellucida (ZP) and corona radiata (CR) was suspended in the antrum. Theca interna (TI) and theca externa (TE) were clearly differentiated in the mature follicle. The corpus luteum (CL) consisted of granulosa lutein (GL) cells and small areas of theca lutein (TL) cells. Blood vessels (BV) were clearly visible in the CL, and a CT capsule enclosed the CL [Fig 7A–7E]. The mean count of secondary (antral) follicles per ovary, the mean count of corpora lutea, the mean thickness (μm) of granulosa layer, the mean thickness (μm) of theca folliculi, and the mean count of degenerated (atretic) follicles were presented in [Fig 8A–8F].

The ***P. dioscoridis group*** showed a normal histological structure of the ovary similar to the control group [Fig 7F]. There was no significant difference between the control and *P. dioscoridis* groups regarding the mean count of secondary

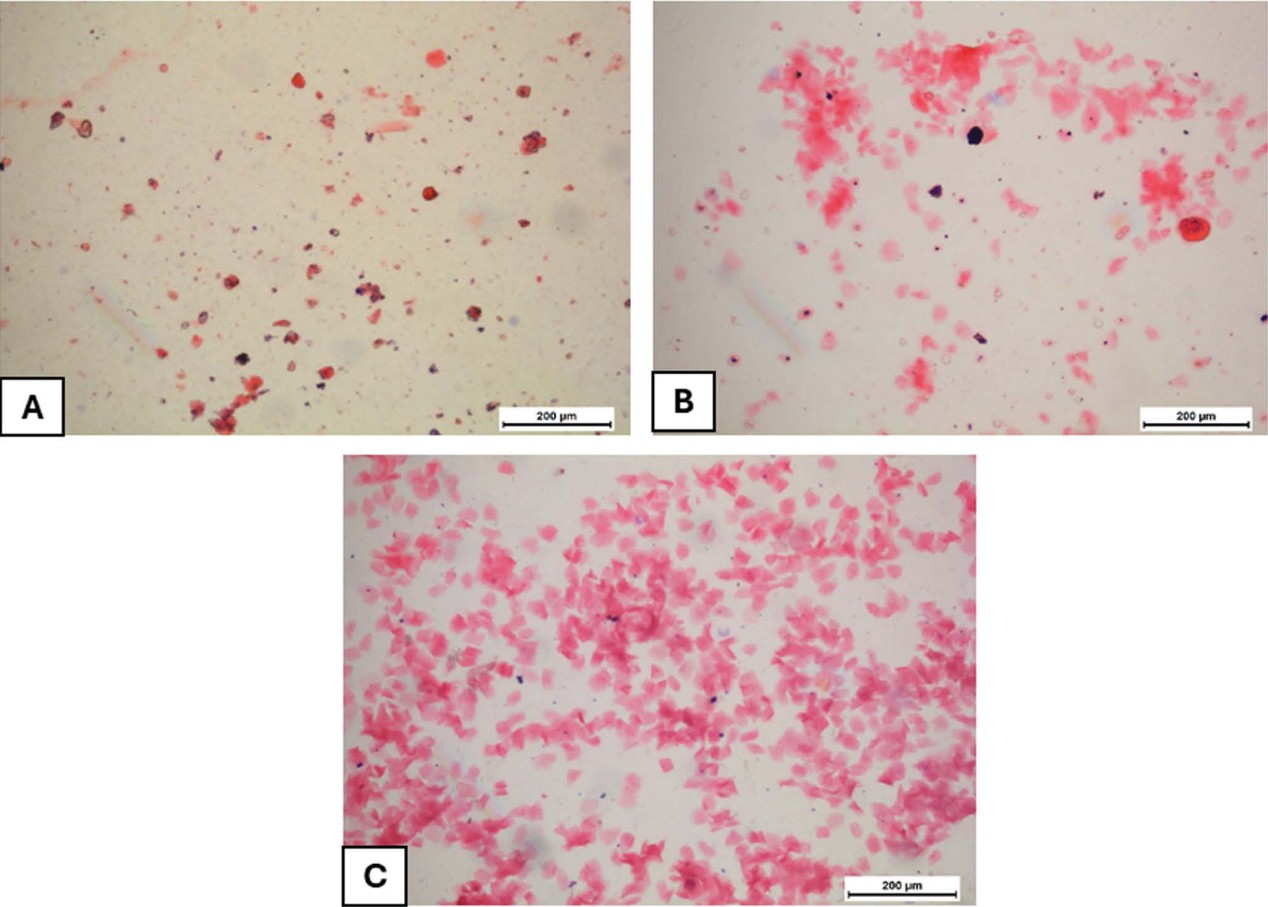

**Fig 4. Vaginal smear at different stages of estrous cycle after PCOS induction and treatment.** (A) PCOS group with diestrous phase. (B) Metformin treated group with metestrous phase. (C) P. dioscoridis treated group with estrous phase.

follicles, the mean count of corpora lutea, the mean count of degenerated follicles, the mean thickness (μm) of the granulosa layer, and the mean thickness (μm) of theca folliculi [Fig 8A–8F].

The **PCOS group** had large cystic follicles (CF) in the ovarian cortex that occupied the entire cortex. Few secondary (antral) follicles or corpora lutea (CL) were detected. Many degenerated follicles were observed compared to the control group [Fig 9A, 9B]. A cystic follicle (CF) had a thin wall surrounding large cavity (antrum) of liquor folliculi. The wall had thick a theca externa (TE) and a thin granulosa (G) layer in comparison to the control group [Fig 9A, 9B]. Some of the cystic follicles (CF) had sloughing of granulosa cells in the antrum There was a significant (P ≤ 0.05) decline in the mean count of secondary (antral) follicles per ovary and the mean count of corpora lutea compared to the control group [Fig 8C, 8D]. The mean count of degenerated (atretic) follicles and the mean count of cystic follicles increased significantly (P ≤ 0.05) compared to the control group [Fig 8E,8F]. The mean thickness (μm) of the granulosa layer decreased significantly (P ≤ 0.05) compared to the control group, while the mean thickness (μm) of the theca folliculi increased significantly (P ≤ 0.05) in comparison to the control group [Fig 8A, 8B].

The **PCOS + metformin group** showed improvement in the histological structure of the ovary which revealed fewer cystic follicles (CF) in the ovarian cortex compared to the PCOS group. Some secondary (antral) follicles were observed in the cortex [Fig 9C]. The mean thickness (μm) of the granulosa layer decreased significantly (P ≤ 0.05) in comparison to the

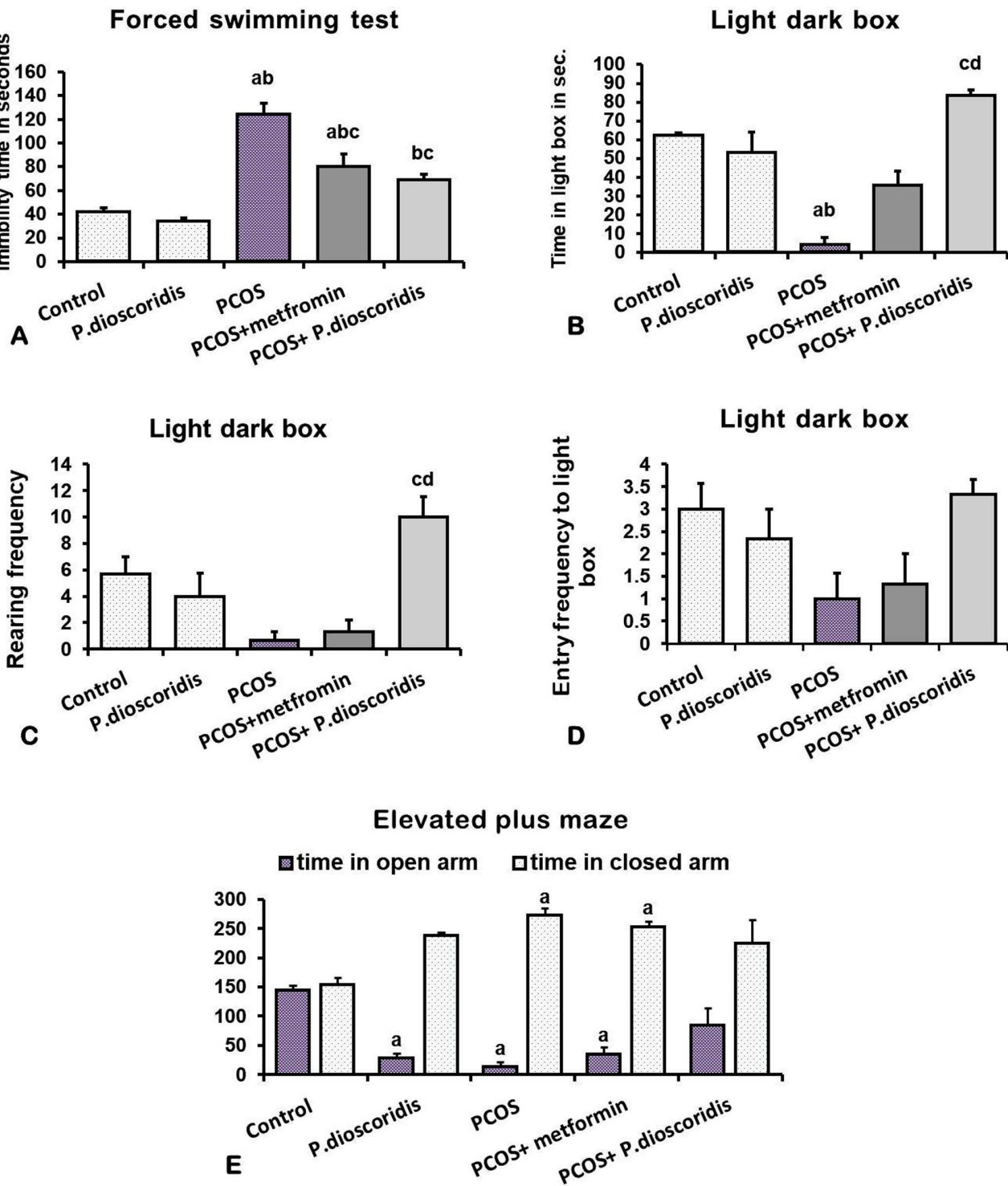

**Fig 5. Effect of metformin and P. dioscoridis on anxiety and depressive like behaviors in letrozole-induced PCOS..** Data are represented by **Means ± SE.** All data were analyzed using ANOVA followed by Bonferroni post hoc test. a (significant versus CTRL group), b (significant versus *P. dioscoridis* group), and c (significant versus PCOS).

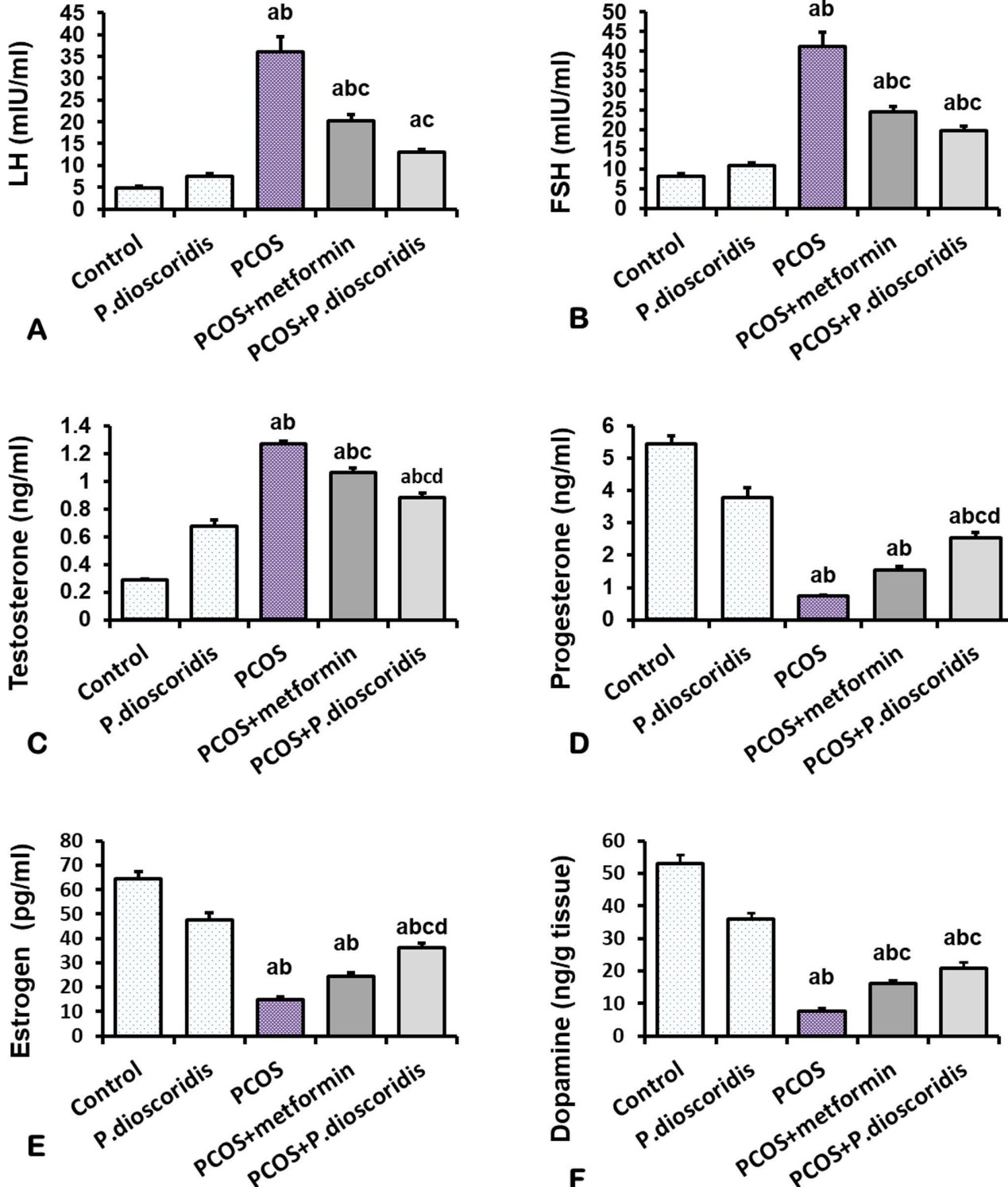

**Fig 6. Effect of metformin and *P. dioscoridis* on sex hormonal profile and brain dopamine level in letrozole-induced PCOS.** Data are represented by Means ± SE. All data were analyzed using ANOVA followed by Bonferroni post hoc test. a (significant versus CTRL group), b (significant versus *P. dioscoridis* group), c (significant versus PCOS), and d (significant versus metformin+ PCOS group).

**Table 3. Pearson correlation between brain dopamine levels and behavioral analysis.**

| | | Immobility time in Forced swimming test | Time in Closed arm (elevated plus maze) | Time in open arm (elevated plus maze) |
|---|---|---|---|---|
| Dopamine | Pearson correlation | −.817** | −.737** | .677** |
| | Sig. (2-tailed) | .000 | .002 | 0.006 |

**Table 4. Pearson correlation between brain dopamine levels and hormonal results.**

| | | Testosterone | LH | FSH | Progesterone | Estrogen |
|---|---|---|---|---|---|---|
| **Dopamine** | *Pearson correlation* | −.971** | −.833** | −.858** | .964** | .958** |
| | *Sig. (2-tailed)* | .000 | .000 | .000 | .000 | .000 |

control, while the mean thickness (μm) of the theca folliculi declined significantly (P ≤ 0.05) compared to the PCOS group [Fig 8A, 8B]. The mean count of cystic follicles and atretic follicles declined significantly compared to the PCOS group [Fig 8E, 8F]. There was no significant difference regarding the mean count of secondary (antral) follicles or corpora lutea versus the PCOS group [Fig 8B, 8C].

The **PCOS + P. dioscoridis group** showed marked improvement in the histological structure of the ovary compared to PCOS group [Fig 9D, 9E]. There was marked (P ≤ 0.05) improvement in the thickness of the granulosa and theca compared to the PCOS group [Fig 8A, 8B]. There was a marked (P ≤ 0.05) decline in the count of cystic and atretic follicles and a significant (P ≤ 0.05) increment in the count of corpora lutea in comparison to the PCOS group [Fig 8D–8F].

**3.9.2. Masson's trichrome stain.** The **control group** showed greenish collagen fibers in the connective tissue (CT) stroma of the cortex and medulla, in the wall of blood vessels (BV), in the fibrous theca externa (TE) of ovarian follicles, and in the CT capsule and core of corpora lutea (CL) [Fig 10A]. The mean color area % of greenish collagen in the ovary was presented in [Fig 10F].

The **P. dioscoridis group** showed normal collagen fiber deposition in the ovary similar to the control group [Fig 10B]. The mean color area % of greenish collagen in the ovary was presented in [Fig 10F].

The **PCOS group** showed increased collagen fibers deposition in the ovary in the CT stroma of the cortex and medulla. There was also increased collage fiber deposition in the thickened fibrous theca externa (TE) ensheathing the cystic follicles (CF) [Fig 10C]. The mean color area % of greenish collagen in the ovary increased significantly (P ≤ 0.05) compared to the control group [Fig 10F].

The **PCOS + metformin group** showed collagen fiber deposition in the ovary similar to that of the PCOS group [Fig 10D]. There was no significant difference between the PCOS + metformin group and the PCOS group regarding the mean color area % of greenish collagen [Fig 9F].

The **PCOS + P. dioscoridis group** showed decreased collagen fiber deposition in the ovary compared to the PCOS group [Fig 10E]. The mean color area % of greenish collagen in the ovary decreased significantly (P ≤ 0.05) compared to the PCOS group [Fig 10F].

**3.9.3. Ki 67 immunostaining.** The **control group** showed ki67 immunoreaction in the granulosa (G) cells of the ovarian follicles, some granulosa lutein (GL) cells, and theca lutein (TL) cells of the corpus luteum (CL). The immunoreaction appeared as dark brown nuclear staining [Fig 11A]. The mean color area % of ki67 immunoreaction in the ovary was presented in [Fig 11F].

The **P. dioscoridis group** showed ki67 immunoreaction in the ovary similar to that of the control group [Fig 11B]. The mean color area % of ki67 immunoreaction in the ovary was presented in [Fig 11F].

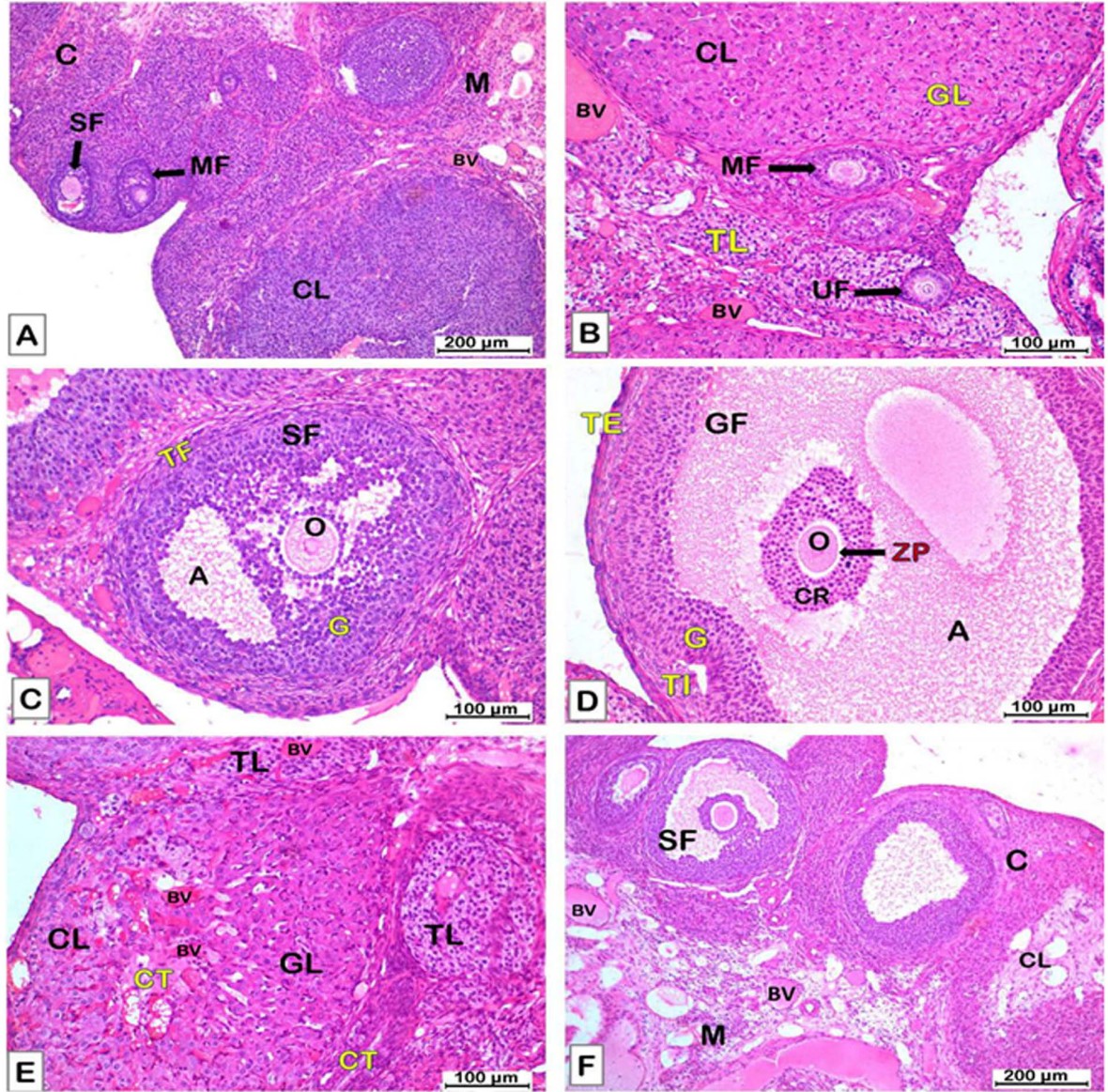

**Fig 7. A photomicrograph of sections of the ovary in the control and P. dioscoridis groups [H&E x100 (A, F), H&E x200 (B-E)].** A) The control group [H&E x 100] shows normal architecture of the ovary with superficial cortex (C) and deep medulla (M). The cortex has various ovarian follicles. The medulla has loose connective tissue rich in blood vessels (BV). Multilaminar primary follicle (MF), secondary follicle (SF), and corpus luteum (CL) are seen in the cortex. B) A higher magnification of the previous section [H&E x 200] shows unilaminar (UF) and multilaminar (MF) primary follicles. UF is formed of a single layer of cuboidal follicular cells surrounding an oocyte. MF is formed of several rows of cuboidal granulosa cells enclosing an oocyte. A corpus luteum (CL) with small pale theca lutein cells (TL), and large acidophilic granulosa lutein cells (GL) is visible in the section. Blood vessels (BV) are clearly seen. C) A secondary follicle (SF) [H&E x 200] with antral (A) cavity and granulosa (G) cells surrounding an oocyte (O). A layer of theca folliculi (TF) envelopes the secondary follicle. D) Mature Graffian follicle (GF) [H&E x 200] has large antrum (A) contains liquor folliculi surrounded with membrana granulosa (G), theca interna (TI), and theca externa (TE). An oocyte (O) surrounded with zona pellucida (ZP) and corona radiate (CR) is suspended in the antrum. E) A corpus luteum (CL) [H&E x 200] with large granulosa lutein (GL) cells and small theca lutein (TL) cells enclosed within connective tissue capsule (CT), and has a core of CT and is richly vascularized (BV) is seen in the cortex. F) The P. dioscoridis group shows normal structure of the ovary similar to that of the control group [H&E x 100].

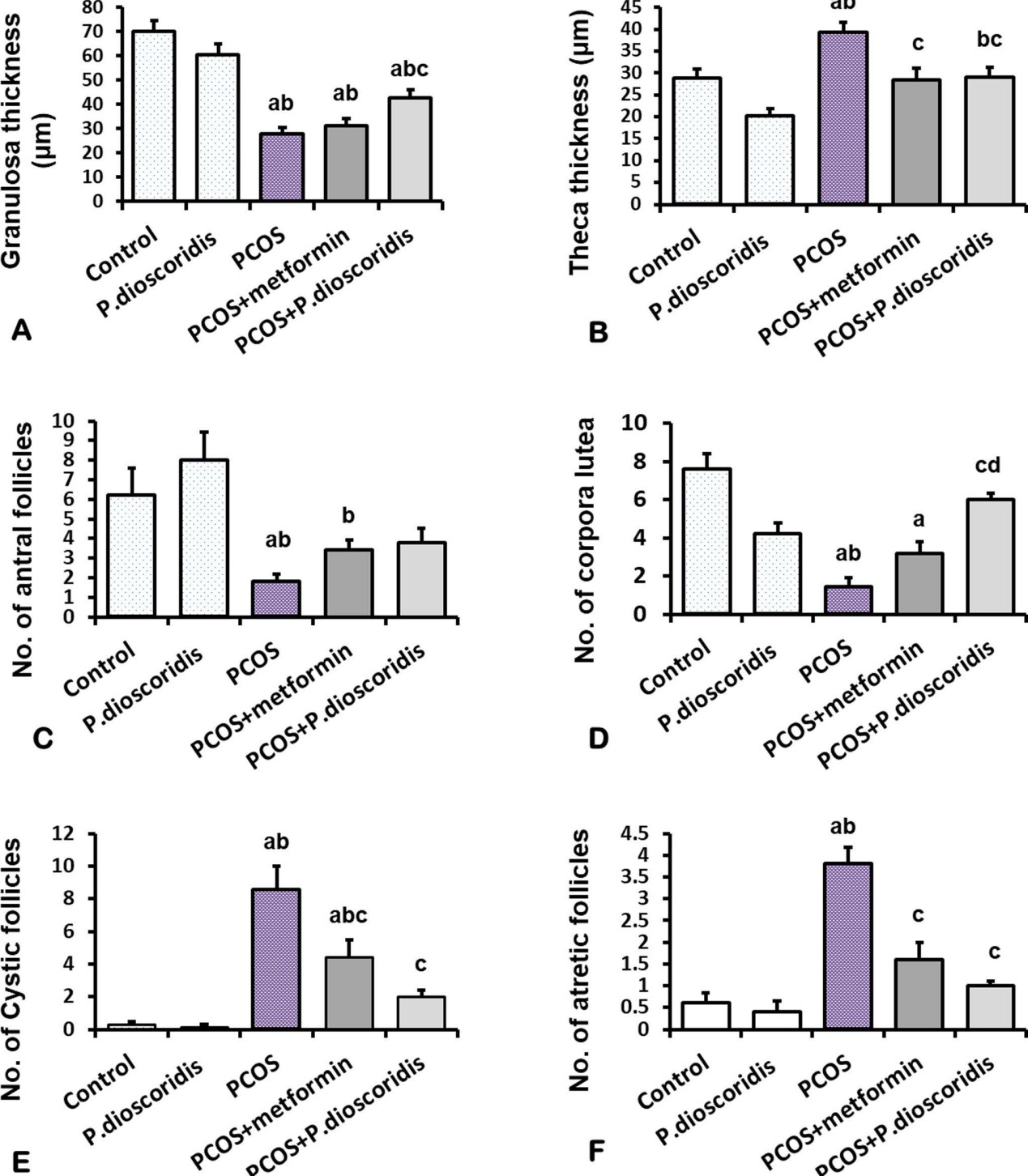

**Fig 8. A comparison among study groups regarding: A) Granulosa thickness (µm), B) Theca thickness (µm), C) Number of antral follicles, D) Number of corpora lutea, E) Number of cystic follicles, F) Number of atretic follicles.** Data are analyzed by one-way ANOVA and Bonferroni post hoc test at $P < 0.05$ and presented as mean ±SE. [a] versus the control group, [b] versus the *P. dioscoridis* group, [c] versus the PCOS group, and [d] versus the PCOS+metformin group.

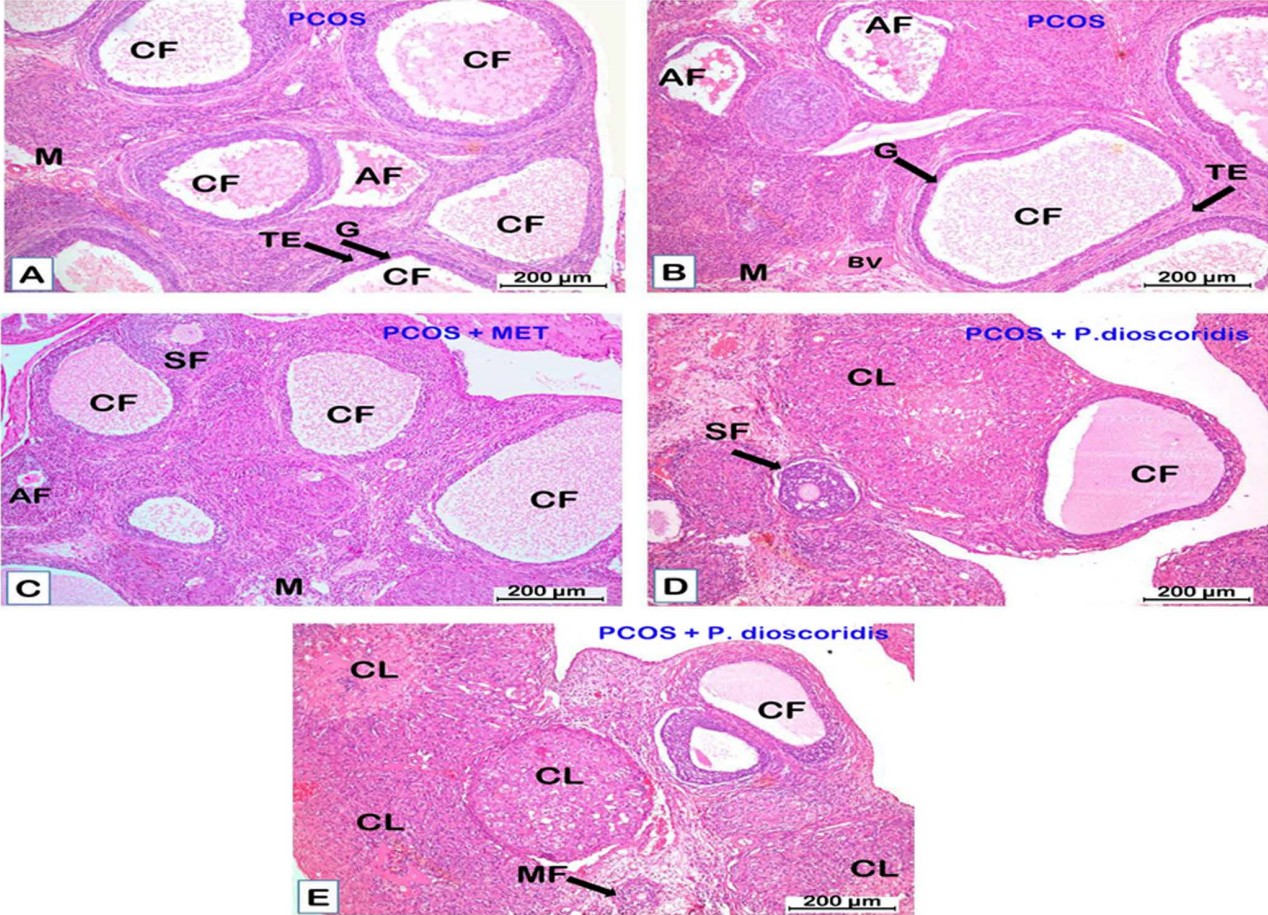

**Fig 9. A photomicrograph of sections of the ovary in the PCOS, PCOS +metformin, and PCOS+*P. dioscoridis* groups [H&E x100].** A-B). The PCOS group shows large cystic follicles (CF) in the ovarian cortex with thin granulosa (G) layer and thick theca externa (TE) layer compared to the control group. Many atretic (degenerated) ovarian follicles (AF) are seen in the cortex. C) PCOS+metformin group shows fewer and smaller cystic follicles (CF) in the ovarian cortex compared to that observed in PCOS group. A secondary follicle (SF) and an atretic follicle (AF) also could be detected. D-E) PCOS+*P. dioscoridis* group shows many corpora lutea (CL) in the ovarian cortex. Multilaminar primary follicle (MF) and secondary follicle (SF) are also present. Few cystic follicles (CF) could be detected in the cortex.

The **PCOS group** showed decreased ki67 immunoreactive nuclei of the granulosa cell layer compared to the control group [Fig 11C]. The mean color area % of ki67 immunoreaction in the ovary decreased significantly (P ≤ 0.05) compared to the control group [Fig 11F].

The **PCOS+metformin group** showed ki67 immunoreaction in the ovary nearly similar to the PCOS group [Fig 11D]. The mean color area % of ki67 immunoreaction in the ovary declined significantly (P ≤ 0.05) compared to the control group [Fig 11F].

The **PCOS+*P. dioscoridis* group** showed an increased ki67 immunoreactive nuclei in the ovary compared to the PCOS group [Fig 11E]. The mean color area % of ki67 immunoreactivity in the ovary increased significantly (P ≤ 0.05) compared to the PCOS group [Fig 11F]. There was a significant (P ≤ 0.05) difference between PCOS + metformin and PCOS+*P. dioscoridis* groups regarding the mean color area % of ki67 immunoreaction [Fig 11F].

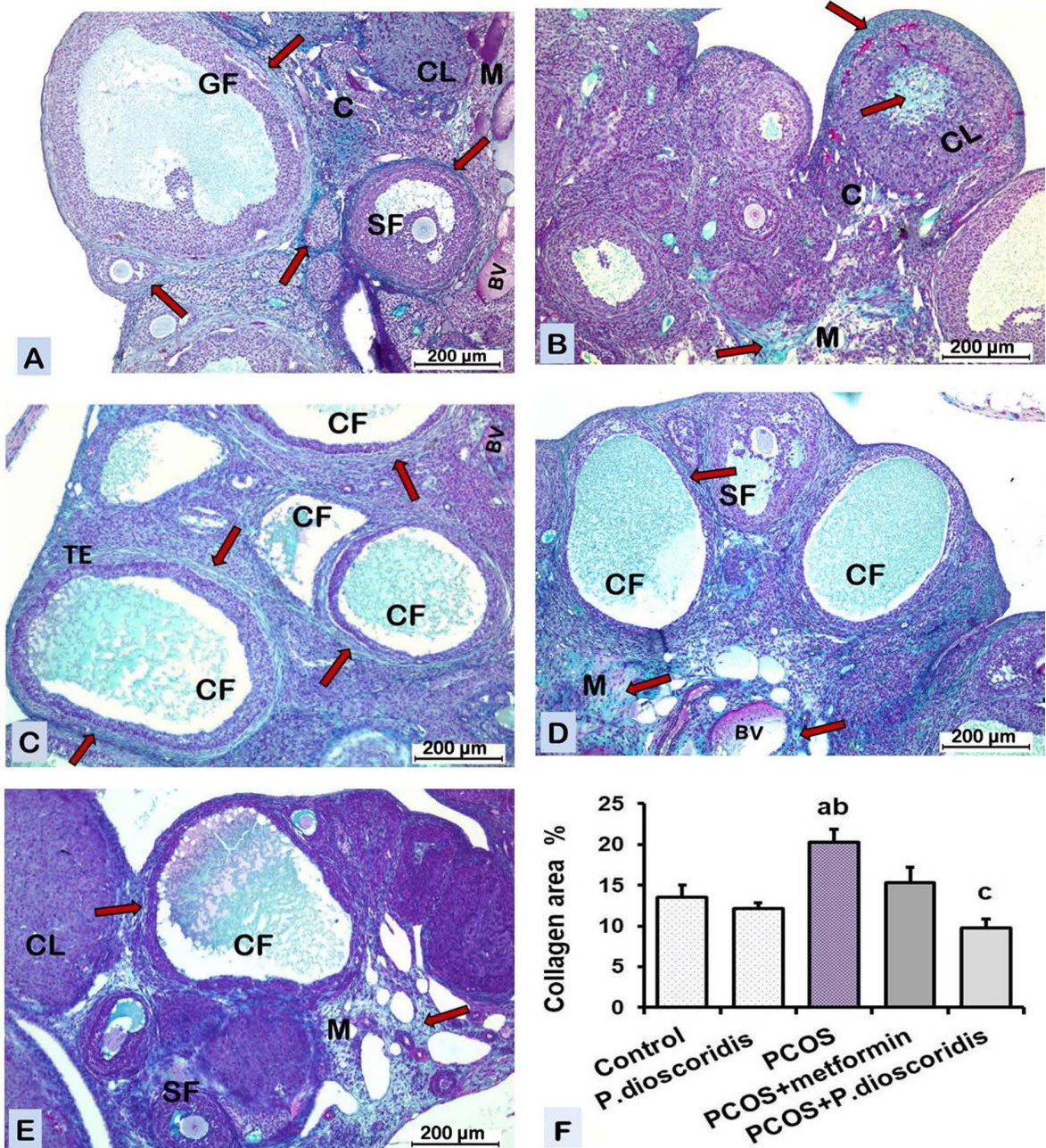

**Fig 10. A photomicrograph of sections of the ovary in different groups [Masson's trichrome x100].** A-B) The control and P. dioscoridis groups respectively showing greenish collagen fibers (arrow) in the stroma of the cortex (C), medulla (M), the theca externa (arrow) ensheathing the ovarian follicles, the connective tissue of the corpus luteum (CL). C) The PCOS group shows increased collagen fibers (arrow) deposition in the stroma of the cortex and medulla and the theca exterrna (TE) encircling the cystic follicles (CF). D) PCOS+metformin group shows collagen fibers (arrow) deposition in the stroma of cortex and medulla, ensheathing cystic follicles (CF) and blood vessels (BV) nearly similar to PCOS group. E) PCOS+P. dioscoridis group shows decreased collagen fibers deposition in the ovary compared to PCOS group. F) Chart represents color area percentage of greenish collagen fibers in different groups. Data are analyzed by one-way ANOVA and Bonferroni post hoc test at P<0.05 and presented as mean ±SE. [a] versus the control group, [b] versus the P. dioscoridis group, and [c] versus the PCOS group.

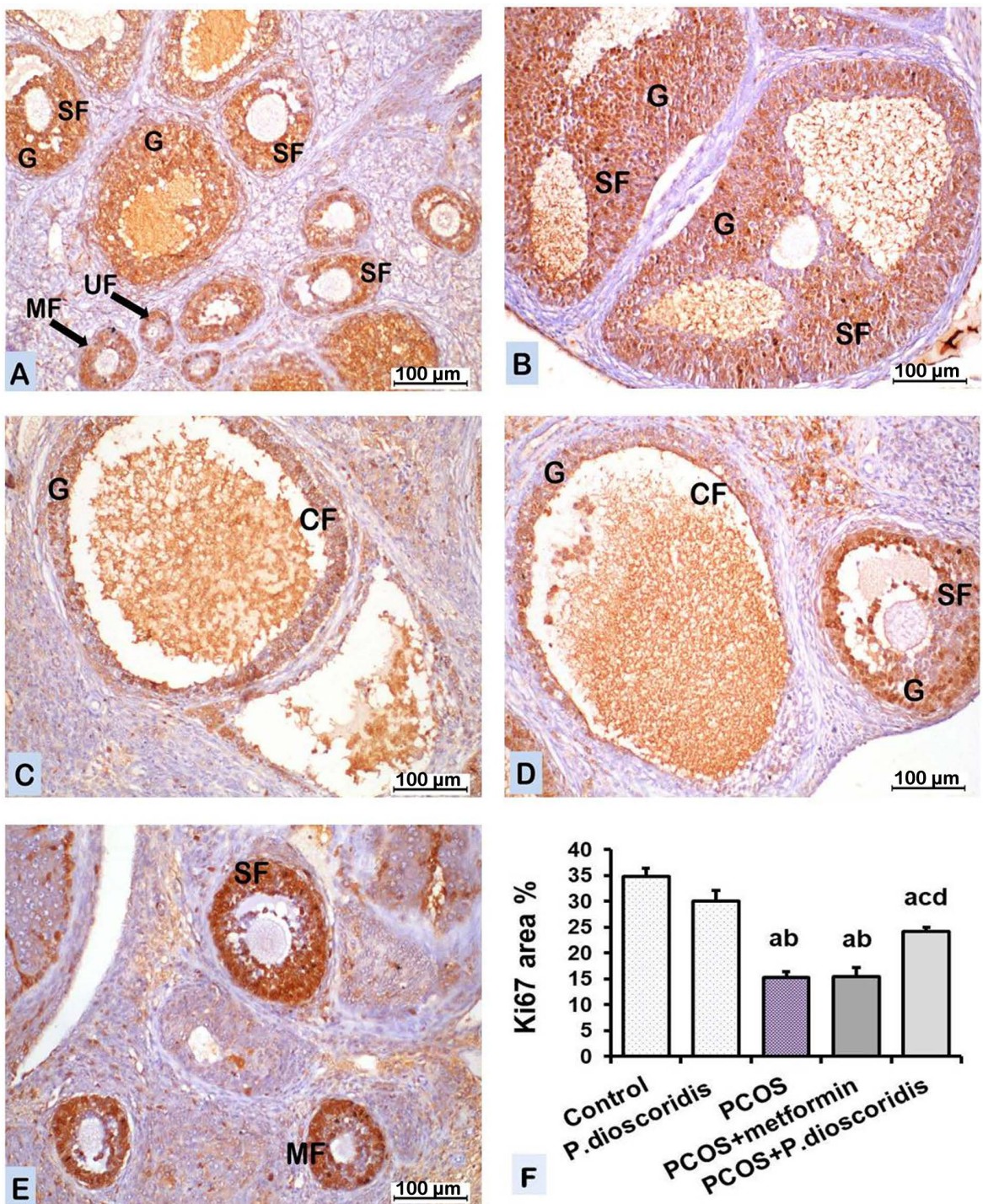

**Fig 11. A photomicrograph of sections of the ovary in different groups [ki67 immunostaining x200].** A) The control group shows ki67 immunore-action in the granulosa (G) cells of different ovarian follicles, the unilaminar primary (UF), the multilaminar primary (MF), and the secondary (SF) follicle. The reaction appears as a dark brown nuclear staining. B) The *P. dioscoridis* group shows ki67 immunoreaction in the granulosa (G) cells of ovarian follicles similar to control group. C) The PCOS group shows decreased ki67 positive cells compared to control group. D) PCOS+metformin group shows decreased ki67 positive cells compared to control group. It is nearly similar to PCOS group. E) PCOS+*P. dioscoridis* group shows increased ki67 positive cells compared to PCOS group. F) Chart represents color area percentage of ki67 immunoreaction in different groups. Data are analyzed by one-way ANOVA and Bonferroni post hoc test at P<0.05 and presented as mean ±SE. [a] versus the control group, [b] versus the *P. dioscoridis* group, [c] versus the PCOS group, and [d] versus the PCOS+metformin group.

**Table 5. Effect of metformin and *P. dioscoridis* ethanolic extract on dopamine receptor 2, DOPA decarboxylase and MAO-A gene expressions in letrozole-induced PCOS.**

| Experimental group | Dopamine receptor 2 | DOPA Decarboxylase | MOA-A |
|---|---|---|---|
| Control group | $1 \pm 0$ | $1 \pm 0$ | $1 \pm 0$ |
| *P. dioscoridis* group | $2.07 \pm 0.83$ | $1.52 \pm 0.26$ | $0.7 \pm 0.08$ |
| PCOS group | $0.24 \pm 0.08$ | $0.31 \pm 0.04$ | $2.45 \pm 0.61^{ab}$ |
| PCOS + Metformin group | $3.71 \pm 0.29^{ac}$ | $2.92 \pm 0.51^{c}$ | $0.07 \pm 0.01^{c}$ |
| PCOS + *P. dioscoridis* group | $5.18 \pm 0.31^{abc}$ | $4.54 \pm 0.86^{abc}$ | $0.06 \pm 0.02^{c}$ |

Data are represented by Means ± SE. All data were analyzed using ANOVA followed by Bonferroni post hoc test. a (significant versus CTRL group), b (significant versus *P. dioscoridis* group), and c (significant versus PCOS).

### 3.10. Effect of metformin and *P. dioscoridis* extract on D$_2$R gene expression

There was decreased D$_2$R expression in letrozole induced PCOS group compared to control group Both metformin and *P. dioscoridis* extract treated groups showed significantly elevated D$_2$R gene expression compared to the PCOS and control groups ($P < 0.001$) [Table 5].

### 3.11. Effect of metformin and *P. dioscoridis* extract on DOPA decarboxylase gene expression

Both metformin and *P. dioscoridis* extract-treated groups showed significantly elevated dopamine decarboxylase gene expression compared to the PCOS and control groups ($P < 0.001$) [Table 5].

### 3.12. Effect of metformin and *P. dioscoridis* extract on MAO-A gene expression

There were significantly high levels of MAO-A gene expression in letrozole induced PCOS group compared to the control group ($P < 0.001$). Both metformin and *P. dioscoridis* extract-treated groups showed significantly decreased MAO-A expression compared to the PCOS group ($P < 0.001$) [Table 5].

## 4. Discussion

The research demonstrated that *P. dioscoridis* alcohol extract possesses a diverse chemical composition with 56 metabolites, which include *26 flavonoids, 5 fatty acids, and 8 phenolic acid derivatives.* It exhibited substantial therapeutic benefits for PCOS, surpassing metformin across multiple parameters. Both medications restored estrous cycle regularity and ameliorated depressive-like behavior and anxiety, with *P. dioscoridis* showing effects that were comparable to the control group. *P. dioscoridis* was more efficacious in regulating sex hormone profiles by diminishing LH, FSH, and testosterone levels while augmenting progesterone and estrogen. It also markedly enhanced brain dopamine levels and demonstrated a positive correlation with improved hormonal and behavioral results. Histological examinations demonstrated that *P. dioscoridis* reestablished ovarian architecture, diminished cystic and deteriorated follicles, and augmented granulosa layer thickness, accompanied by reduced collagen deposition and heightened Ki67 immunostaining, signifying cellular regeneration. Furthermore, *P. dioscoridis* enhanced the expression of dopamine receptor 2 and dopamine decarboxylase genes while diminishing MAO-A expression, hence augmenting its remarkable anxiolytic and neuroprotective properties.

### 4.1. Chemical components of *P. dioscoridis* EtOH extract'

Numerous studies have reported that various extracts of *P. dioscoridis* exhibit a wide range of potent biological activities, including anti-inflammatory, antiulcer, antidiabetic, a, and free radical scavenging effects [16]. Seven hydroxycinnamic acid esters, including *O*-caffeoyl quinic acid derivatives, were identified in *P. dioscoridis* samples, highlighting the plant's notable antioxidant potential (peaks 6, 12, 16, 20, 21, 23, and 24) [38]. Additionally, flavonoids found include luteolin (tetrahydroxyflavone),

quercetin, jaceidin and tricin, Quercetin and luteolin are among the most common plant-derived compounds, widely distributed across the plant kingdom and regularly consumed in the human diet. While their potential role in preventing neurodegenerative and lifestyle-related diseases remains under discussion, recent studies—including clinical trials—have provided growing evidence in support of their benefits [38]. Nitrogenous compounds were found in *P. dioscoridis including* sphingosines, xanthurenic and pantothenic (Vitamin B5) acids. Vitamin B5, for example, is crucial for the synthesis of coenzyme A (CoA), which functions as a carbonyl activator and acyl group carrier, supporting numerous biochemical reactions [39].

### 4.2. Flavonoids

The 176, 162, 146, and 132 Da product ions, which correspond to the presence of pentose, deoxyhexose, hexose, and hexuronyl moieties, correspondingly, could result from the cleavage of the *O*-glycosidic link that connects the flavonoid aglycone to the glycoside unit [46]. Conversely, *C*-hexosides of the flavonoid revealed fragmentations at −150, −120, and −90 Da corresponding to the partial cleavagings of the sugar however, *C*-pentosides of flavonoid exhibited loss neutrally at −60 Da. Also, RDA-F (retro Diels-Alder fission) is another documented way for the flavonoid aglycones fragmentation [48].

Peak number 10 demonstrated parent ion peak at the *m/z* 565.1553 [M+H]$^+$ supporting a formula $C_{26}H_{29}O_{14}{}^+$ and exhibited daughter ions at *m/z* 415 [M+H-150]$^+$, 445 [M+H-120]$^+$, 475 [M+H-90]$^+$, and 505 [M+H-60]$^+$ corresponding to *C*-hexosyl and *C*-pentosyl moieties. On the other hand, the fragmentations at *m/z* 385 [M+H-180]$^+$ as well as the *m/z* 356 [M+H-209]$^+$ owing to the aglycone connected to sugar moieties, i.e., [apigenin (270)] + (115) and (86), respectively demonstrated the di-*C*-substituted derivative of flavone. Thus, peak 10 was assigned as isoschaftoside (apigenin-*C*-pentosyl-*C*-hexoside) [48] and first detected in the *P. dioscoridis* genus. In the same context, peaks 11 and 13 exhibited molecular ions at the m/z 449.1074 [M+H]$^+$, $C_{21}H_{21}O_{11}{}^+$ and 433.1133 [M+H]$^+$, $C_{21}H_{21}O_{10}{}^+$, respectively and daughter ions at both of [M+H-120]$^+$ along with [M+H-150]$^+$ due to the presence of *C*-hexosyl groups. Therefore, peaks 11 and 13 were annotated as luteolin-*C*-hexoside [46] and isovitexin, respectively, reported for the first time in *P. dioscoridis*. Peaks 8 and 18 had the same molecular ion at *m/z* 595.166 [M+H]$^+$, $C_{27}H_{31}O_{15}{}^+$ albeit, different fragmentations at m/z 475 [M+H-120]$^+$ and 287 [M+H-146–162]$^+$ concluded the identification of kaempferol-*C*-deoxyhexosyl-*C*-hexoside and kaempferol-*O*-deoxyhexosyl-hexoside, respectively [46]. Trihydroxyflavone-*O*-dihexoside was the compound's designation after peak 27 (m/z 593.1528 [M-H]$^-$, $C_{27}H_{29}O_{15}{}^-$) showed the successive losses of the two *O*-hexosyl units at *m/z* 431 [M-H-162]$^-$ and 269 [M-H-2×162]$^-$ [46]. Peak 16 (m/z 465.1025 [M+H]$^+$, $C_{21}H_{21}O_{12}{}^+$) also showed the identical fragmentation pattern thought to be caused by the loss of the *O*-hexosyl group [M+H-162]$^+$, and this was determined to be isoquercitrin (quercetin-*O*-hexoside). Peak 17 showed a fragmentation fingerprint of the eliminated hexuronyl moiety (−176 Da), which was identified as kaempferol-*O*-glucuronide.

At m/z 360, 342, and 317, peak 38 (m/z 375.1061 [M+H]$^+$, $C_{19}H_{19}O_8{}^+$) exhibited distinctive daughter ions that were ascribed to [M+H-methyl]+, [M+H-methyl-$H_2O$]+, and [M+H-2methyl-CO]$^+$, respectively. According to data that has already been published, peak 38's annotation was verified as casticin (dihydroxy-tetramethoxyflavone) [54]. Peak 41 (m/z 315.0856 [M+H]$^+$, $C_{17}H_{15}O_6{}^+$) showed fragmentation ions at m/z 300 [M+H-methyl]$^+$, 282 [M+H-methyl-$H_2O$]$^+$, and 257 [M+H-2methyl-CO]$^+$, which corresponded to a polymethoxylated flavonoid, i.e., luteolin 7,3-dimethyl ether (velutin), similar to casticin in terms of its fragmentation pattern [55]. Both velutin and casticin were detected herein for the first time in *P. dioscoridis* genus. Peak 29 revealed a fragment ion at m/z 315 [M+H-162]$^+$ owing to the presence of *O*-hexosyl moiety and was identified as cirsimarin. Likewise, peaks 14 and 15 revealed similar fragmentation patterns with a mass difference of 30 Da ($OCH_2$) and were assigned as rutin (pentahydroxyflavone-*O*-deoxyhexosyl-hexoside) and pentahydroxy-methoxyflavone-*O*-deoxyhexosyl-hexoside, respectively [46]. Other flavonoid aglycones were detected on negative ion mode in peaks 30 (m/z 285.0415 [M-H]$^-$, $C_{15}H_9O_6{}^-$), 31 (m/z 301.0356 [M-H]$^-$, $C_{15}H_9O_7{}^-$), 33 (m/z 359.076 [M-H]$^-$, $C_{18}H_{15}O_8{}^-$) and 37 (m/z 329.0663 [M-H]$^-$, $C_{17}H_{13}O_7{}^-$) corresponding to luteolin (tetrahydroxyflavone), quercetin, jaceidin, and tricin, respectively.

## 4.3. Phenolic acids

Having potent antioxidant activity [61], seven hydroxy cinnamic acid esters were detected in *P. dioscoridis* samples for the first time. The *P. dioscoridis* is known for its *O*-caffeoyl quinic acid esters, which were identified in peaks 6, 12, 16, 20, 21, 23, and 24 [62]. The *O*-caffeoyl quinic acid was designated as peak 6 (*m/z* 353.0874 [M-H]-, $C_{16}H_{17}O_9^-$), which was identified through the loss of the caffeoyl group at *m/z* 191 [M-H-162]-, which was ascribed to deprotonated quinic acid. The protonated caffeoyl moiety at m/z 193 and the neutralizing degradation of quinic acid at *m/z* 177 [M+H-192]$^+$ allowed daughter ions to be observed in another less polar peak, number 12 (*m/z* 369.1184 [M+H]$^+$, $C_{17}H_{21}O_9^+$), validating *O*-caffeoylquinic acid methyl ester. Due to the sequential disappearance of two caffeoyl fragments (−2×162), peaks 20 and 23 displayed daughter ions, which were determined to be dicaffeoylquinic acid and its isomer. The identical fragment pattern appeared in peak 24 (*m/z* 499.1231 [M+H]$^+$, $C_{25}H_{23}O_{11}^+$), that was recognized as dicaffeoylquinic acid lactone following two successive losses of caffeoyl fragments. In addition, peak 22 (*m/z* 449.1076 [M+H]$^+$, $C_{21}H_{21}O_{11}^+$) generated a daughter ion at m/z 193 [M+H-194-$H_2O$-$CO_2$]$^+$ and a base peak at m/z 255 [M+H-194]$^+$ presumably a result of neutrality degradation of ferulic acid. Similarly, a fragment ion with lower relative abundance was found at *m/z* 273 [M+H-176]$^+$ as a result of the neutral loss of the feruloyl group, indicating that this component was cimicifugic acid [51].

## 4.4. Nitrogenous compounds

In certain primary peaks, even molecular ions that were ascribed to the existence of nitrogenous atoms were detected. Consecutive loss of water was observed at m/z 202 [M+H-18]$^+$ and 184 [M+H-36]$^+$ in peak 4 (*m/z* 220.1183 [M+H]$^+$, $C_9H_{18}NO_5^+$). Additionally, the breakdown of amidic and Cα–CO bonds generated protonated *β*-alanine and an acyl ion, respectively, resulting in two fragmentation peaks at m/z 60 and 116. Consequently, pantothenic acid was determined to peak 4. In addition to fragmentation at m/z 252 [M+H-18–30]$^+$ ascribed to the disappearance of formaldehyde and one water moieties, peak 44 (*m/z* 300.2887 [M+H]$^+$, $C_{18}H_{38}NO_2^+$) showed successive losses of water at m/z 282 and 264. As a result, peak 44 was identified as $C_{18}$-sphingosine. Peak 5 was identified as xanthurenic acid and showed daughter ions at m/z 178 [M+H-CO]$^+$, 160 [M+H-CO-$H_2O$]$^+$, and 132 [M+H-CO-$H_2O$-CO]$^+$ (*m/z* 206.0446 [M+H]$^+$, $C_{10}H_8NO_4^+$). Alongside sphingosines, xanthurenic and pantothenic (Vitamin B5) acids were found for the first time in *P. dioscoridis*. In addition to the two distinctive daughter ions at m/z 169 [M+H-$C_8H_{12}N_2$]$^+$ and 153 [M+H-$C_4H_9O_2PS$]$^+$, peak 38 (*m/z* 305.1071 [M+H]$^+$, $C_{12}H_{22}N_2O_3PS^+$) also showed daughter ions at m/z 277 [M+H-28]$^+$ and 249 [M+H-2×28]$^+$, which were ascribed to the slow disappearance of ethylene moieties. Diazinon was the annotation name for this metabolite. Daughter ions were formed at m/z 436 [M+H-$H_2O$]$^+$, 393 [M+H-$H_2O$-$C_2H_5N$]$^+$, and 313 [M+H-141]$^+$ due to the depletion of phosphoethanolamine, and 216 [M+H-238]$^+$ due to the degradation of the palmitoyl moiety via peak 49 (*m/z* 454.2925 [M+H]$^+$ $C_{21}H_{45}NO_7P^+$). Palmitoyl-glycero-phosphoethanolamine, which was originally observed in *P. dioscoridis*, was identified as peak 38. First identified in *P. dioscoridis*, peaks 53 (*m/z* 324.2888 [M+H]$^+$, $C_{18}H_{34}NO^+$) and 55 (*m/z* 338.3406 [M+H]$^+$, $C_{22}H_{44}NO^+$) were identified as linoleamide and docos-enamide (erucamide). Comparable fragmentation patterns were observed in both metabolites, which were ascribed to the loss of water [M+H-18]$^+$ and ammonia [M+H-17]$^+$. Vitamin B5 is crucial for the synthesis of coenzyme A (CoA), a key molecule in fatty acid metabolism and energy production. It plays an important role in the biosynthesis of fatty acids, cholesterol, steroid hormones, amino acids, and neurotransmitters [39].

## 4.5. The dual role of dopamine in behavioral regulation and hormonal balance

This research offers significant insights into the correlation among brain dopamine levels, behavioral effects, and hormone regulation. The findings indicate a significant negative correlation between dopamine levels and immobility time in the forced swimming test, implying that dopamine is crucial in mitigating depressive-like behavior. Dopamine levels exhibited a reverse correlation with the duration spent in the closed arm of the elevated plus maze, suggesting its function in alleviating anxiety, but a positive correlation with open-arm exploration reinforces dopamine's role in fostering exploration and mitigating anxiety. The study reveals a significant relationship between dopamine and reproductive hormones: dopamine levels are strongly negatively

correlated with testosterone and positively correlated with LH, FSH, progesterone, and estrogen, indicating dopamine's essential role in regulating the hypothalamic-pituitary-gonadal axis. These findings highlight dopamine's dual function in regulating behavioral states and reproductive health, providing a more profound comprehension of its intricate physiological roles.

Sexual steroids regulate neuroendocrine diencephalic regions like the hypothalamus in the brain. Steroids also affect cortical, limbic, and midbrain regions, affecting memory, learning, mood, and reward [63]. In concordance with our results, a study highlighted the interaction between ovarian hormones and dopamine pathways in regulating the behavioral inhibition system (BIS) and the behavioral activation system (BAS). Estradiol and progesterone showed a combined effect on BIS scores, with estradiol positively or negatively correlated depending on progesterone levels. High progesterone levels enhance the substantia nigra-dorsolateral prefrontal cortex connection, according to resting-state functional connectivity investigations. These findings show hormonal regulation of dopamine-related motivation and inhibition [64].

## 4.6. The comparative effect of metformin and *P. dioscoridis* on hormonal modulation in PCOS

Metformin improves androgen, estrogen, and progesterone levels in this study, however, *P. dioscoridis* extract gives more significant results. The effect of metformin may be attributed to its insulin sensitivity improvement in PCOS [65]. On the other hand, this search demonstrated the therapeutic potential of *P. dioscoridis* on hormonal disturbance in letrozole-induced PCOS rats. Despite that the pathogenesis of PCOS is still limited, Yuan et al., 2025 demonstrated a link between PCOS pathogenesis and hormonal fluctuations in the levels of LH, FSH, androgens, progesterone, and estrogen [66]. Flavonoids, being one of the components of *P. dioscoridis* have been shown to lower LH and testosterone levels in PCOS rat models through modulating the IL-6-mediated JAK2\STAT3 signaling pathway [67]. Quercetin, a flavonoid found in *P. dioscoridis* extract, increases CYP19A1 and CYP11A1 expressions leading to elevating estrogen levels in letrazole-induced PCOS mice model [68], in addition to improving insulin resistance through its antioxidant properties [69]. Caffeic acid a phenolic acid derivative of *P. dioscoridis*, alleviates endoplasmic reticulum stress and inhibits 3β-HSD protein in DHEA-induced PCOS rat model [70]. All these multi-target properties in *P. dioscoridis* extract give it the privilege over metformin in the treatment of this heterogenous disease.

**The comparative effect of metformin and *P. dioscoridis* on ovarian morphological alterations**Histological analysis revealed that the PCOS + metformin group exhibited fewer cystic and atretic follicles and a thinner theca than the PCOS group. Similar findings have been reported in earlier research [33,71,72].

In the PCOS + *P. dioscoridis* group, histological analysis revealed reestablished ovarian architecture, diminished cystic and deteriorated follicles, and augmented granulosa layer thickness, accompanied by reduced collagen deposition and heightened Ki67 immunostaining, signifying cellular regeneration. According to earlier studies, flavonoids and phenolic acids included in *P. dioscoridis* metabolites have a role in mitigating polycystic ovary syndrome [73,70]. Quercetin lowers the activity of the enzyme that converts progesterone to androgens by reducing the expression of the 17α-hydroxylase/C17-20-lyase (CYP17) gene. This relieves PCOS's excessive androgen and anovulation conditions and returns oocytes to normal meiosis [73]. Additionally, quercetin increases the gene for the estrogen receptor α (Erα), which is crucial for fertility [74]. Caffeic acid, was shown in another study by Chiang et al. (2023) to have antioxidative and anti-inflammatory properties and to improve PCOS by preventing oxidative stress and apoptosis [70]. Caffeic acid treatment resulted in a recovery of folliculogenesis, a significant increase in corpora lutea count, and an increase in granulosa cell layer thickness, with a reduction in cystic follicles and theca cell layer thickness. Moreover, Samani et al. (2024) detected improved follicular development, decreased number of cysts, increased corpora lutea, and decreased density of collagen deposition after treatment of rats with apigenin [75]. Apigenin is a flavonoid included in the metabolites of *P. dioscoridis* extract. Thus, these findings clarify the ameliorating effects of *P. dioscoridis* ethanolic extract on letrozole-induced PCOS in the current work.

## 4.7. The comparative effect of metformin and *P. dioscoridis* on dopamine pathway

The dopamine production process entails the transformation of tyrosine into L-DOPA via tyrosine hydroxylase, succeeded by DOPA decarboxylase (DDC) turning L-DOPA into dopamine, which is then metabolized by monoamine oxidase (MAO)

[76]. The PCOS group in the current study exhibited considerable downregulation of DOPA decarboxylase, resulting in decreased dopamine synthesis, whereas high MAO-A expression accelerates dopamine degradation, leading to markedly diminished dopamine levels. Metformin treatment considerably augmented the expressions of DOPA decarboxylase, and type 2 dopamine receptors, thereby improving dopamine production and sensitivity, while simultaneously decreasing MAO-A expression, which reduced dopamine breakdown, thereby returning dopamine levels to near average. *P. dioscoridis* independently enhanced DOPA decarboxylase expression and type 2 dopamine receptor, while moderately suppressing MAO-A, demonstrating a beneficial effect on dopamine modulation. This indicates that the treatments target critical regulatory sites in dopamine synthesis, sensitivity, and degradation, and sheds light on the possibility of the combined therapy to exhibit the most effective restoration of dopamine homeostasis impaired by PCOS. In concordance with our results, Ng et al., 2012 metformin treatment has been shown to be neuroprotective in Parkinson's disease (PD), reducing neurotoxicity and alleviating dopaminergic dysfunction [77]. However, in a study using rats, metformin failed to protect dopaminergic neurons in response to intranigral lipopolysaccharide injection [78].

### 4.8. The potential mitigating effects of metformin and *P. dioscoridis* on depression and anxiety

Metformin improved anxiety and depression-related disorder, however, its effect was less pronounced than *P. dioscoridis.* The therapeutic potential of metformin may be attributed to its antioxidant property since Kakhki et al, 2024 linked the antidepressive and anxiolytic properties of metformin to the attenuation of neuroinflammation and oxidative stress induced by lipopolysaccharide-induced inflammation [79].

A systematic review analyzed the correlation between dietary polyphenols and depression, utilizing data from 163 animal studies, 16 observational studies, and 44 intervention studies. Animal research consistently demonstrated that polyphenols mitigate depression-like behaviors, however, human studies present inconclusive outcomes, with some suggesting a protective impact and others identifying no correlation. Further investigation is required, particularly in younger and clinical human cohorts [80]. Ferulic acid, a phenolic molecule included in *P. dioscoridis*, demonstrated considerable antidepressant potential by inhibiting monoamine oxidase (especially MAO-A) and enhancing monoamine neurotransmitters. In male ICR mice, ferulic acid (20–80 mg/kg) diminished immobility in depression assessments and increased brain serotonin (5-HT) and norepinephrine (NE) concentrations, indicating its potential to mitigate depressive-like behaviors [81].

Previous research validated the anxiolytic properties of natural small-molecule phenols (NSMPs), evaluated by the elevated plus maze in murine subjects. Seven phenolic compounds, including ferulic acid, eugenol, and caffeic acid, demonstrated anxiolytic effects. The phenolic hydroxyl group of 4-hydroxycinnamic acid (4-OH CA) was essential for its activity, evidenced by its superior effects relative to 4-chlorocinnamic acid (4-Cl CA) in behavioral assessments and hippocampal spike recordings. Findings indicate that the hippocampus CA1 area participates in the anxiolytic route of 4-OH CA. The findings underscore the therapeutic potential of NSMPs and their widespread occurrence in herbal remedies [82]. A study assessed Quercetin in *P. dioscoridis*, and its protective effects against immobilization stress in mice. Behavioral tests and brain analysis showed Quercetin reversed stress-induced anxiety, depression, and cognitive impairment, improved memory, reduced lipid peroxidation, and restored antioxidant enzyme activity. Quercetin also increased acetylcholine levels, decreased acetylcholinesterase, and enhanced serotonin metabolism in stressed mice, indicating its potential as an antianxiety, antidepressant, and memory-enhancing agent [83]. Among the biological constituents found in *P. dioscoridis* in our study is apigenin. Apigenin flavone, a natural bioactive compound, is widely studied for its therapeutic potential. Key constituents include glycosylated apigenin, vitexin, apiin, isovitexin, and rhoifolin. Among these, vitexin, a *C*-glycosylated apigenin, is gaining attention for its multi-modal actions and benefits in various diseases. Vitexin also inhibits the monoamine oxidase B (MAO-B) enzyme, increasing striatal dopamine levels and improving behavioral deficits in experimental Parkinson's disease models [84]. All the aforementioned are supportive for our findings in the current study in relation to behavioral results caused by *P. dioscoridis.*

Fig 12 summarizes the mechanisms behind modulation of neurobehavioral dysfunctions, ovarian abnormalities, and dopamine levels of *P. dioscoridis* on letrozole-induced PCOS. These findings lead us to hypothesize that *P. dioscoridis*

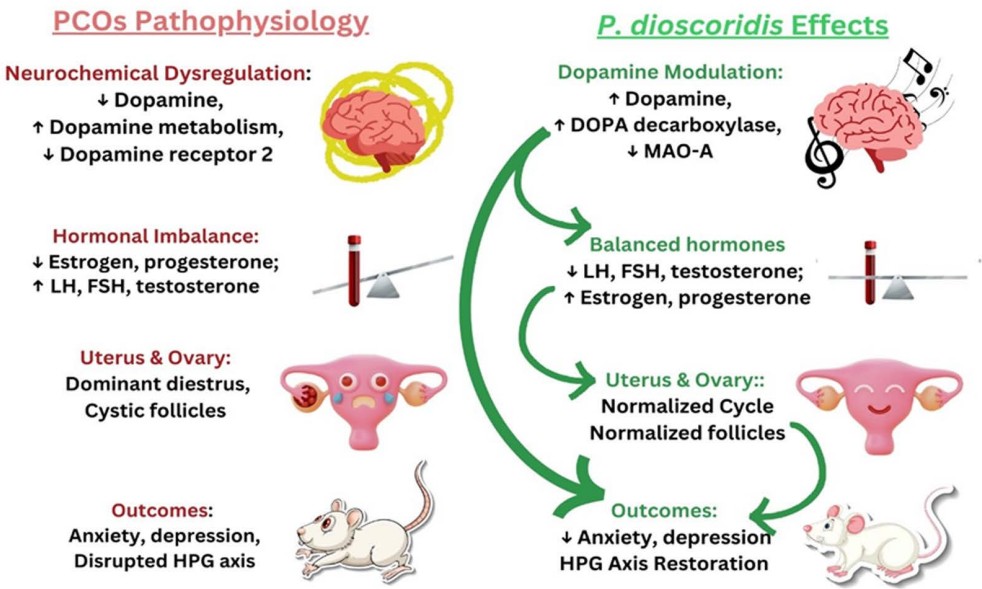

**Fig 12. Hypothesized P. dioscoridis Mechanism in PCOS.** The pathophysiology of PCOS may be counteracted by P. dioscoridis, as shown in the figure. In addition to hormonal imbalances (decreased estrogen/progesterone, increased LH/FSH/testosterone), PCOS is characterized by neurochemical dysregulation with decreased dopamine and dopamine receptor activity and ovarian abnormalities like cystic follicles. These alterations are linked to depression, anxiety, and disturbances of the HPG axis. P. dioscoridis restores hormone levels, enhances ovarian structure, reduces neurobehavioral disorders, and may restore dopamine levels by upregulating DOPA decarboxylase and blocking MAO-A.

treatment may be able to reduce ovarian dysfunction, depression and anxiety associated with letrozole-induced PCOS.

### 4.9. Conclusion and future perspectives

This study emphasizes the diverse therapeutic potential of *P. dioscoridis* and its bioactive components, especially in regulating dopaminergic pathways. The combined effects of *P. dioscoridis* and metformin exhibit significant efficacy in reestablishing dopamine homeostasis, positioning them as attractive options for managing disorders including PCOS-induced dopaminergic dysfunction and dopamine-imbalance-related diseases. Moreover, the anxiolytic and antidepressant effects of *P. dioscoridis*, reinforced by its flavonoid components such as quercetin and vitexin, highlight its potential to mitigate stress-induced behavioral and cognitive deficits. The results highlight the plant's function in modulating monoamine neurotransmitters, enhancing antioxidant defenses, and maintaining hormonal equilibrium, so establishing a robust basis for its application in formulating innovative treatment approaches for neurodegenerative and neuropsychiatric conditions. Subsequent studies must concentrate on clinical validation and the deeper clarification of molecular pathways to reinforce its therapeutic applicability.

### Acknowledgments

The authors appreciatively acknowledge technical support in animal house care, Faculty of Medicine, Suez Canal University, Egypt.

## Author contributions

**Conceptualization:** Rasha Atta, Mohamed A. Farag, Shimaa Mohammad Yousof.

**Data curation:** Rasha Atta.

**Formal analysis:** Rasha Atta.

**Funding acquisition:** Thamer Alqurashi, Shimaa Mohammad Yousof.

**Investigation:** Rasha Atta, Sahar Galal Gouda, Marwa Hussein Mohamed, Sherif M. Afifi, Mahmoud I. Nassar, Abdelsamed I. Elshamy, Shimaa Mohammad Yousof.

**Methodology:** Rasha Atta, Sahar Galal Gouda, Marwa Hussein Mohamed, Sherif M. Afifi, Mahmoud I. Nassar, Abdelsamed I. Elshamy, Thamer Alqurashi, Shimaa Mohammad Yousof.

**Project administration:** Rasha Atta.

**Software:** Rasha Atta, Sahar Galal Gouda, Marwa Hussein Mohamed, Sherif M. Afifi, Mahmoud I. Nassar, Abdelsamed I. Elshamy, Shimaa Mohammad Yousof.

**Supervision:** Rasha Atta, Abdelsamed I. Elshamy, Shimaa Mohammad Yousof.

**Validation:** Rasha Atta.

**Visualization:** Rasha Atta, Mohamed A. Farag.

**Writing – original draft:** Rasha Atta, Sahar Galal Gouda, Marwa Hussein Mohamed, Sherif M. Afifi, Mahmoud I. Nassar, Abdelsamed I. Elshamy, Mohamed A. Zayed, Shimaa Mohammad Yousof.

**Writing – review & editing:** Rasha Atta, Mohamed A. Farag, Abdelsamed I. Elshamy, Shimaa Mohammad Yousof.

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
