## [Decision Letter · Decision Letter 0]

PONE-D-25-08188

Pluchea dioscoridis Extract: A Novel Therapeutic Approach for Polycystic Ovary Syndrome Targeting Ovarian Morphology, Dopamine Pathways, and Neurobehavior in Relation to its Phytocomponents

Dear Dr. Atta,

Thank you for submitting your manuscript to PLOS ONE. After careful consideration, we feel that it has merit but does not fully meet PLOS ONE’s publication criteria as it currently stands. Therefore, we invite you to submit a revised version of the manuscript that addresses the points raised during the review process.

We look forward to receiving your revised manuscript.

Kind regards,

Sanaz Alaeejahromi

Academic Editor

PLOS ONE

**Comments to the Author**

1. Is the manuscript technically sound, and do the data support the conclusions?

Reviewer #1: Yes

Reviewer #2: Yes

2. Has the statistical analysis been performed appropriately and rigorously?

Reviewer #1: Yes

Reviewer #2: Yes

3. Have the authors made all data underlying the findings in their manuscript fully available?

Reviewer #1: Yes

Reviewer #2: Yes

4. Is the manuscript presented in an intelligible fashion and written in standard English?

Reviewer #1: No

Reviewer #2: Yes

5. Review Comments to the Author

Reviewer #1: In the abstract, it is mentioned that: all the treatments were for 21 days. It seems that it should be revised since after 21 days of letrozole for PCOS induction, metformin and P. dioscoridiswas used.

The result section of the abstract must be written in a way to shows the significance or non-significant of changes statistically.

In the Introduction add a reference for this sentence: Animal models of depression, including the learned helplessness and chronic mild stress (CMS) models, demonstrate substantial alterations in the dopamine (DA) system.

Also, add a reference for this: This encompasses modified dopamine receptor expression in limbic

areas, diminished synaptic dopamine release, lowered levels of the dopamine metabolite homovanillic acid, and decreased dopaminergic activity in the striatum.

The below studies are suggested:

The chronic mild stress (CMS) model of depression: History, evaluation and usage

Method:

For this section add reference: The air-dried plant materials (820 gm) were macerated in 4 liters of a 7:3 mixture of ethanol and distilled water for 5 days at the room temperature then filtered. After

completing three rounds of this process, all the liquor extracts were gathered and dried under vacuum, producing 34.2 g of black gum that was kept refrigerated at 4 °C until additional testing was done.

The below study is suggested:

Capacity of Mentha spicata (spearmint) Extract in Alleviating Hormonal and Folliculogenesis Disturbances in Polycystic Ovarian Syndrome Rat Model

Add the weight of rats.

Add a reference for the selection of dose of letrozole. In most of the studies 28 days of letrozole administration is used. Such as the below study. It is suggested to explain about the dose selection using studies such as below:

Blood volatile organic compounds as potential biomarkers for poly cystic ovarian syndrome (PCOS): An animal study in the PCOS rat model

Also use other studies which used metformin or P. dioscoridis and explain about the criteria for selecting similar or different doses in comparison to other studies.

For example, in the below study metformin 500 for 1 week was used instead of 300 for 21 days.

The effects of melatonin and metformin on histological characteristics of the ovary and uterus in letrozole-induced polycystic ovarian syndrome mice: A stereological study

In this section mention the temperature or method for preservation for subsequent studies: Both brain and ovaries were collected for further biochemical and histopathological studies.

If the authors established the method for Ki 67 immunostaining, it is suggested to add its reference.

If the primers are designed by the authors it is suggested to mention the software or if they were selected from the studies add reference.

Results

Delete repetitive numbers in the result section if they are presented in figures or tables.

Reviewer #2: Title

The title is comprehensive.

Abstract

The abstract provides a clear overview of the study's objectives, methods, results, and conclusions.

The introduction is well structured.

The methods section is well-structured.

some figures (e.g., Figure 5) could be improved for clarity, particularly in labeling and the presentation of statistical significance. Additionally, the results section could benefit from a more detailed discussion of the chemical profile of P. dioscoridis and how specific phytocomponents may contribute to the observed effects.

The discussion is comprehensive and effectively interprets the results in the context of existing literature. It highlights the potential mechanisms by which P. dioscoridis exerts its therapeutic effects, particularly in relation to dopamine modulation and hormonal balance. The discussion also appropriately addresses the limitations of the study, such as the need for further clinical validation. However, the section could be more focused in some areas, particularly in the discussion of the chemical components of P. dioscoridis. While the identification of flavonoids and phenolic acids is mentioned, the discussion could delve deeper into how these specific compounds might interact with biological pathways relevant to PCOS. Additionally, the discussion could benefit from a more critical evaluation of the comparative efficacy of P. dioscoridis versus metformin, particularly in light of the study's findings.

---

## [Author Response · Author response to Decision Letter 1]

4 May 2025

we submitted a revised version of the manuscript that addresses the points raised during the review process.

---

## [Decision Letter · Decision Letter 1]

Pluchea dioscoridis Extract: A Novel Therapeutic Approach for Polycystic Ovary Syndrome Targeting Ovarian Morphology, Dopamine Pathways, and Neurobehavior in Relation to its Phytocomponents

PONE-D-25-08188R1

Dear Dr. Rasha Atta,

We’re pleased to inform you that your manuscript has been judged scientifically suitable for publication and will be formally accepted for publication once it meets all outstanding technical requirements.

Kind regards,

Sanaz Alaeejahromi

Academic Editor

PLOS ONE

Reviewer #1: All comments have been addressed

Reviewer #2: All comments have been addressed

2. Is the manuscript technically sound, and do the data support the conclusions?

Reviewer #1: Yes

Reviewer #2: Yes

3. Has the statistical analysis been performed appropriately and rigorously? 

Reviewer #1: I Don't Know

Reviewer #2: Yes

4. Have the authors made all data underlying the findings in their manuscript fully available?

Reviewer #1: Yes

Reviewer #2: Yes

5. Is the manuscript presented in an intelligible fashion and written in standard English?

Reviewer #1: Yes

Reviewer #2: Yes

6. Review Comments to the Author

Reviewer #1: Thank you for your comprehensive response to all comments and for your commitment to strengthening this valuable contribution to the field.

Reviewer #2: The authors have demonstrated exceptional responsiveness in addressing reviewer comments reflecting a strong commitment to manuscript improvement

---

## [Editor Report · Acceptance letter]

PONE-D-25-08188R1

PLOS ONE

Dear Dr. Atta,

I'm pleased to inform you that your manuscript has been deemed suitable for publication in PLOS ONE. Congratulations! Your manuscript is now being handed over to our production team.

Kind regards,

on behalf of

Dr. Sanaz Alaeejahromi

Academic Editor

PLOS ONE